# An oligomeric state-dependent switch in the ER enzyme FICD regulates AMPylation and deAMPylation of BiP

Luke A Perera[1] iD, Claudia Rato[1] iD, Yahui Yan[1] iD, Lisa Neidhardt[1] iD, Stephen H McLaughlin[2] iD, Randy J Read[1] iD, Steffen Preissler[1,*] iD & David Ron[1,**] iD

## Abstract

AMPylation is an inactivating modification that alters the activity of the major endoplasmic reticulum (ER) chaperone BiP to match the burden of unfolded proteins. A single ER-localised Fic protein, FICD (HYPE), catalyses both AMPylation and deAMPylation of BiP. However, the basis for the switch in FICD's activity is unknown. We report on the transition of FICD from a dimeric enzyme, that deAMPylates BiP, to a monomer with potent AMPylation activity. Mutations in the dimer interface, or of residues along an inhibitory pathway linking the dimer interface to the enzyme's active site, favour BiP AMPylation *in vitro* and in cells. Mechanistically, monomerisation relieves a repressive effect allosterically propagated from the dimer interface to the inhibitory Glu234, thereby permitting AMPylation-competent binding of MgATP. Moreover, a reciprocal signal, propagated from the nucleotide-binding site, provides a mechanism for coupling the oligomeric state and enzymatic activity of FICD to the energy status of the ER.

**Keywords** AMPylation; BiP; deAMPylation; endoplasmic reticulum; FICD
**Subject Categories** Translation & Protein Quality; Structural Biology
**The EMBO Journal (2019) 38: e102177**

## Introduction

In all domains of life, protein folding homeostasis is achieved by balancing the burden of unfolded proteins and the complement of chaperones. In the endoplasmic reticulum (ER) of animal cells, this match is facilitated by the unfolded protein response (UPR). In addition to well-recognised transcriptional and translational strands of the UPR (Walter & Ron, 2011), recent findings have drawn attention to the existence of rapid post-translational mechanisms that adjust the activity of the ER Hsp70 chaperone BiP (Preissler & Ron, 2019). Best understood amongst these is AMPylation, the covalent addition of an AMP moiety from ATP onto a hydroxyl group-containing amino acid side chain.

AMPylation conspicuously occurs on Thr518 of BiP (Preissler *et al*, 2015b; Broncel *et al*, 2016; Casey *et al*, 2017). The resulting AMPylated BiP (BiP-AMP) is heavily biased towards a domain-coupled ATP-like state (Preissler *et al*, 2015b, 2017b; Wieteska *et al*, 2017). Consequently, BiP-AMP has high rates of client protein dissociation (Preissler *et al*, 2015b). Moreover, the ATPase activity of BiP-AMP is resistant to stimulation by J-domain protein co-factors, which greatly reduces the chaperone's ability to form high-affinity complexes with its substrates (Preissler *et al*, 2017b). AMPylation therefore serves to inactivate BiP. This modification is temporally dynamic and the levels of BiP-AMP respond to changes in ER protein folding load (Preissler *et al*, 2015b).

Consistent with its inactivating character, BiP modification in cells is enhanced by inhibition of protein synthesis (Laitusis *et al*, 1999) or during recovery from ER stress, when BiP levels exceed the requirements of unfolded client proteins (Preissler *et al*, 2015b). Conversely, as levels of ER stress increase, the modification is reversed by deAMPylation, recruiting BiP back into the chaperone cycle (Laitusis *et al*, 1999; Chambers *et al*, 2012; Preissler *et al*, 2015b). Accordingly, BiP modification creates a readily accessible pool of latent folding capacity that buffers both ER stress (through deAMPylation) and over-chaperoning (through AMPylation). These features may contribute to the observation in the *Drosophila* visual system, whereby loss of the ability to AMPylate BiP results in light-induced blindness (Rahman *et al*, 2012; Moehlman *et al*, 2018).

AMPylation of BiP is mediated by the ER-localised enzyme FICD (filamentation induced by cAMP domain protein, also known as HYPE) (Ham *et al*, 2014; Preissler *et al*, 2015b; Sanyal *et al*, 2015). FICD is the only known metazoan representative of a large family of bacterial Fic domain proteins (Khater & Mohanty, 2015a). Fic proteins contain a conserved active site motif, HPFx(D/E)GN(G/K)R_1xxR_2, and many possess a glutamate-containing inhibitory alpha helix ($\alpha_{inh}$) responsible for auto-inhibition of their canonical AMPylation activity (Engel *et al*, 2012; Goepfert *et al*, 2013). FICD

1 Cambridge Institute for Medical Research, University of Cambridge, Cambridge, UK
2 MRC Laboratory of Molecular Biology, Cambridge, UK
*Corresponding author. Tel: +44 (0)1223 769 100; E-mail: sp693@cam.ac.uk
**Corresponding author. Tel: +44 (0)1223 768 940; E-mail: dr360@medschl.cam.ac.uk

is a class II Fic protein (with its $\alpha_{inh}$ N-terminal to its Fic domain) and an ER-localised type II, single-pass transmembrane protein, with a short cytoplasmic portion and a large luminal-facing catalytic domain (Worby *et al*, 2009; Bunney *et al*, 2014).

Crystal structures of FICD and other Fic domain proteins suggest that engagement of Glu234 (of the $\alpha_{inh}$) with Arg374 ($R_2$ of the Fic motif) prevents binding of MgATP in a conformation conducive to catalysis (Engel *et al*, 2012; Goepfert *et al*, 2013; Bunney *et al*, 2014; Truttmann *et al*, 2016). Moreover, *in vitro* modification of BiP by purified FICD requires mutation of Glu234, suggesting that an AMPylation repressed state is favoured by wild-type FICD. Remarkably, the Fic domain of FICD is also responsible for BiP deAMPylation: an activity that depends on Glu234 (Casey *et al*, 2017; Preissler *et al*, 2017a) and magnesium (Veyron *et al*, 2019). These findings point to deAMPylation as the default activity of the bifunctional enzyme and implicate Glu234 in a functional switch between the two antagonistic activities of the Fic active site.

The Fic domain of human FICD forms a stable anti-parallel dimer via two dimerisation surfaces (Bunney *et al*, 2014; Truttmann *et al*, 2016) and a monomerising mutation in the dimer interface of *Drosophila* FICD does not prevent BiP deAMPylation *in vitro* (Casey *et al*, 2017). Nonetheless, distantly related bacterial enzymes hint at a possible regulatory role for Fic dimerisation: a mutation in the *Clostridium difficile* Fic (CdFic) dimer interface increased auto-AMPylation (Dedic *et al*, 2016) and changes in oligomeric state affected the activity of the class III Fic protein from *Neisseria meningitidis* (NmFic; Stanger *et al*, 2016).

Here, we report on the biochemical and structural basis of an oligomeric state-dependent switch in FICD's activity, which is well suited to post-translationally regulate protein folding homeostasis in the ER.

## Results

### Disrupting the FICD dimer favours BiP AMPylation

Whilst the *FICD* gene is necessary for BiP AMPylation, overexpression of the wild-type FICD enzyme does not result in a detectable pool of BiP-AMP in cells (Preissler *et al*, 2015b). These findings were explained in terms of dominance of the deAMPylation activity of wild-type FICD, as observed *in vitro* (Preissler *et al*, 2017a). However, somewhere between low-level endogenous expression, which yields physiologically regulated AMPylation, and overexpression, which precludes BiP-AMP accumulation, retrovirally rescued *FICD*$^{-/-}$ cells were endowed with a measure of BiP AMPylation (Figs 1A and EV1A–C). This finding points to a protein-dosage effect on wild-type FICD's activity and suggests that the enzymatic mode of (recombinant) FICD may be affected by its concentration in the ER.

Purified FICD forms a homodimeric complex *in vitro* (Bunney *et al*, 2014). Co-expression of reciprocally tagged FICDs confirmed that the wild-type protein forms homomeric complexes in cells that are disrupted by a previously characterised Leu258Asp mutation within the major dimerisation surface (Bunney *et al*, 2014; Fig 1B). Unlike the wild-type dimerisation-competent enzyme, at a similar level of overexpression, the monomeric FICD$^{L258D}$ yielded a clear BiP-AMP signal in *FICD*$^{-/-}$ cells (Fig 1C). This pool was conspicuous even under basal conditions, in which wild-type cells have only

a weak BiP-AMP signal, suggesting that the imposed monomeric state deregulated FICD's activity.

Together, these observations intimate that dynamic changes in the equilibrium between the monomer and dimer may contribute to a switch between FICD's mutually antagonistic activities—AMPylation and deAMPylation of BiP. Increasing its concentration by over-expression favours FICD dimerisation and thus perturbs such regulatory transitions. This could account for the observation that FICD overexpression, in unstressed wild-type cells, abolishes the small pool of BiP-AMP normally observed under basal conditions (Preissler *et al*, 2017b).

Size-exclusion chromatography (SEC) and analytical ultracentrifugation (AUC), with purified proteins, confirmed the stability of the FICD dimer (Figs 1D and E, and EV1D–G). These techniques also reaffirmed the strong disrupting effect of the Leu258Asp mutation (in the principal dimer surface) and revealed a weaker disrupting effect of a Gly299Ser mutation (in the secondary dimer surface; Fig EV1D–G). AUC yielded a 1.2 nM dimer dissociation constant ($K_d$) of wild-type FICD and SEC indicated a $K_d$ in the millimolar range for FICD$^{L258D}$ and a $K_d$ of 9.5 μM for FICD$^{G299S}$. We therefore conclude that between 0.2 and 5 μM (concentrations at which the experiments that follow were performed), the wild-type protein is dimeric, FICD$^{L258D}$ is monomeric, and FICD$^{G299S}$ is partially monomeric.

In the presence of [$\alpha$-$^{32}$P]-ATP, both FICD$^{L258D}$ and FICD$^{G299S}$ established a pool of AMPylated, radioactive BiP *in vitro* [Fig 1F; also observed in the *Drosophila* counterpart of FICD$^{L258D}$ (Casey *et al*, 2017)], whereas the wild-type enzyme did not, as previously observed (Preissler *et al*, 2015b, 2017a). BiP is a substrate for AMPylation in its monomeric, ATP-bound, domain-docked conformation (Preissler *et al*, 2015b, 2017b). These experiments were therefore performed with an ATPase-deficient, oligomerisation-defective, ATP-bound BiP mutant, BiP$^{T229A-V461F}$. Thus, the BiP-AMP signal is a result of the concentration of substrate (unmodified and modified BiP) and the relative AMPylation and deAMPylation activities of the FICD enzyme. As expected, a strong BiP-AMP signal was elicited by the unrestrained AMPylation-active FICD$^{E234G}$ (which cannot deAMPylate BiP). FICD$^{E234G-L258D}$ gave rise to a similar, but reproducibly slightly weaker, BiP-AMP signal relative to FICD$^{E234G}$.

### Monomerisation switches FICD's enzymatic activities

The ability of the dimer interface FICD mutants to yield a detectable BiP-AMP signal *in vitro* agreed with the *in vivo* data and suggested a substantial change in the regulation of the enzyme's antagonistic activities—either inhibition of deAMPylation, de-repression of AMPylation or a combination of both. To distinguish between these possibilities, we analysed the deAMPylation activities of the FICD mutants in an assay that uncouples deAMPylation from AMPylation. As previously observed, wild-type FICD caused the release of fluorescently labelled AMP from *in vitro* AMPylated BiP, whereas FICD$^{E234G}$ did not (Preissler *et al*, 2017a; Fig 2A). FICD$^{L258D}$ and FICD$^{G299S}$ consistently deAMPylated BiP 2-fold slower than wild-type FICD (Figs 2A and EV2A). The residual *in vitro* deAMPylation activity of FICD$^{L258D}$ and the absence of such activity in FICD$^{E234G}$ are consistent with the divergent effects of expressing these deregulated mutants on a cell-based UPR reporter (Fig EV2B and C).

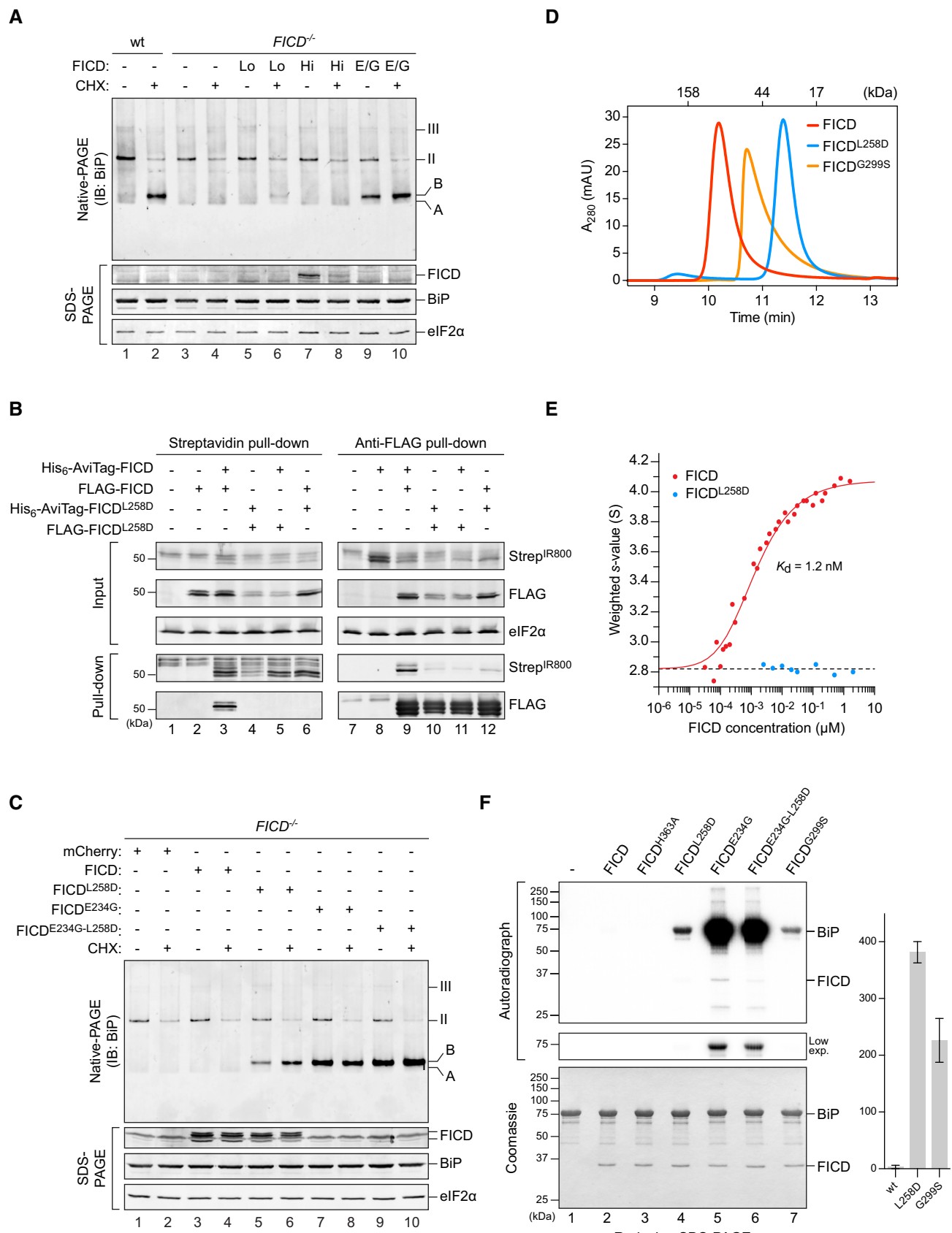

**Figure 1.**

**Figure 1. Monomeric mutant FICD promotes BiP AMPylation.**

A   Immunoblot of endogenous BiP resolved by native-PAGE from lysates of CHO-K1 S21 wild-type (wt) or $FICD^{-/-}$ cells either transiently overexpressing wild-type FICD (high expression level; Hi) or mutant $FICD^{E234G}$ (E/G), or stably expressing recombinant wild-type FICD (low expression level; Lo). The cells in lanes 1–4 were mock-transfected. Where indicated cells were exposed to cycloheximide (CHX; 100 μg/ml) for 3 h before lysis. Unmodified ("A") and AMPylated ("B") monomeric and oligomeric (II and III) forms of BiP are indicated. Immunoblots of the same samples resolved by SDS–PAGE report on FICD, total BiP and eIF2α (loading control). Data representative of four independent experiments are shown. See Fig EV1B and C.

B   Wild-type FICD forms homomeric complexes *in vivo*. Immunoblots of orthogonally tagged wild-type and Leu258Asp mutant FICD in the input cell lysate and following recovery by pull-down with streptavidin (recognising the AviTag) or anti-FLAG antibody. Proteins were detected with fluorescently labelled streptavidin (StrepIR800) or FLAG antibody. Data representative of three independent experiments are shown.

C   Immunoblot of endogenous BiP from transfected CHO-K1 S21 $FICD^{-/-}$ cells (as in A). Note that cells expressing monomeric $FICD^{L258D}$ accumulate AMPylated BiP. Data representative of three independent experiments are shown.

D   Size-exclusion chromatography (SEC) analysis of wild-type and mutant FICD proteins (each at 20 μM). The elution times of protein standards are indicated as a reference. Absorbance at 280 nm ($A_{280\ nm}$) is plotted in units of milli-absorbance units (mAU). Note that the Leu258Asp mutation monomerises FICD, whilst Gly299Ser causes partial monomerisation. See Fig EV1D and E.

E   Comparison of the signal-averaged sedimentation coefficients of wild-type (red) and monomeric mutant $FICD^{L258D}$ (blue), as measured by analytical ultracentrifugation. A fit for monomer-dimer association (solid red line), constrained using the average value for the monomeric protein (dashed line, 2.82 S, $S_{w,20}$ = 3.02 S), yielded a $K_d$ of 1.2 nM with a 95% confidence interval between 1.1 and 1.4 nM and a value of 4.08 S for the dimer ($S_{w,20}$ = 4.36 S). The fitted data points are from three independent experiments. See Fig EV1F and G.

F   Autoradiograph of BiP, AMPylated *in vitro* by the indicated FICD derivatives, with [α-$^{32}$P]-ATP as a substrate and resolved by SDS–PAGE. Proteins in the gel were visualised by Coomassie staining. A representative result of three independent experiments is shown. The graph shows the quantified mean BiP-AMP signals ± SD generated by wild-type FICD and the indicated monomeric mutants.

Source data are available online for this figure.

The FICD-mediated BiP AMPylation/deAMPylation cycle converts the co-substrate ATP to the end products AMP and pyrophosphate (Preissler *et al*, 2017a). We exploited this feature to quantify enzymatic activity. FICD was incubated with [α-$^{32}$P]-ATP, either in the presence or in the absence of ATPase-deficient $BiP^{T229A}$, and accumulation of radioactive AMP was measured by thin-layer chromatography. Only background levels of AMP were generated by catalytically inactive $FICD^{H363A}$ or $FICD^{E234G-H363A}$ (Fig 2B). The deregulated, deAMPylation-defective $FICD^{E234G}$ yielded a weak AMP signal that was not increased further by the presence of BiP, suggesting that the Glu234Gly mutation enables some BiP-independent ATP hydrolysis to AMP. Conversely, small but significant amounts of AMP were produced by wild-type FICD but in a strictly BiP-dependent fashion (Figs 2B and C, and EV2D). These observations are consistent with a slow, FICD-driven progression through the BiP AMPylation/deAMPylation cycle, indicating incomplete repression of wild-type FICD's AMPylation activity under these conditions. As expected, abundant BiP-dependent AMP production was observed in reactions containing AMPylation-active $FICD^{E234G}$ alongside deAMPylation-active wild-type FICD (Fig 2B, lane 11). Importantly, large amounts of AMP were also generated when BiP was exposed to $FICD^{L258D}$ and, to lesser extent, $FICD^{G299S}$ (Figs 2C and EV2D). Together, these observations suggest that the AMPylation activities of the monomeric FICD mutants are significantly enhanced relative to the wild type, whilst their deAMPylation activities are more modestly impaired.

To directly assess the AMPylation activities of bifunctional FICDs, we exploited the high affinity of the catalytically inactive $FICD^{H363A}$ for BiP-AMP, as a "trap" that protects BiP-AMP from deAMPylation (Fig 2D). To disfavour interference with the FICD enzyme being assayed, we engineered the trap as a covalent disulphide-linked dimer incapable of exchanging subunits with the active FICD. A cysteine (Ala252Cys) was introduced into the major dimerisation surface of the trap. To preclude aberrant disulphide bond formation, the single endogenous cysteine of FICD was also replaced (Cys421Ser). After purification and oxidation, the trap ($_{S-S}FICD^{A252C-H363A-C421S}$) formed a stable disulphide-bonded dimer (Fig EV2E and F) that tightly bound BiP-AMP with fast association

and slow dissociation kinetics (Fig EV2G and H). In comparison, binding of the trap to unmodified BiP was negligible (Fig EV2G). We reasoned that adding the trap in excess to reactions assembled with BiP, ATP, and FICD would sequester the BiP-AMP product and prevent its deAMPylation, enabling the direct comparison of AMPylation rates.

In the presence of the trap, wild-type FICD produced a detectable BiP-AMP signal, but not in the absence of the trap (compare Figs 1F and 2E). Importantly, addition of the trap revealed that AMPylation of BiP was greatly accelerated by FICD monomerisation (> 19-fold compared to the wild type) (Fig 2E). As expected, BiP AMPylation by $FICD^{E234G}$ was even faster.

If the enhanced AMPylation activity of the dimerisation-defective mutants, observed above, truly represents divergent enzymatic activities of different FICD oligomeric states, then it should be possible to reveal this feature by diluting the wild-type enzyme to concentrations at which an appreciable pool of monomer emerges. In AMPylation reactions set up with [α-$^{32}$P]-ATP, a detectable signal from radiolabelled BiP-AMP was noted at enzyme concentrations near the $K_d$ of dimerisation (between 10 and 2.5 nM; Fig 3A, left). The inverse correlation between enzyme concentration and the BiP-AMP signal likely reflects the opposing activities and relative populations of AMPylation-biased FICD monomers and the deAMPylation-biased FICD dimers in each reaction. This contrary enzyme-product relationship is resolved in the presence of the AMPylation trap; the BiP-AMP signal increased in a time- and enzyme concentration-dependent manner, as expected from a reaction rate that is proportional to the absolute concentration of monomeric enzyme (Fig 3A, right). In presence of the trap, the shift in the peak of the BiP-AMP signal, after 16 h, towards lower concentrations of FICD, likely reflects incomplete protection of AMPylated BiP by the trap and its enhanced susceptibility to deAMPylation at higher concentrations of (dimeric) FICD.

If monomerisation significantly enhances AMPylation activity, constitutive FICD dimers that are unable to dissociate should have low AMPylation activity and fail to produce modified BiP even under dilute conditions. To test this prediction, we created a

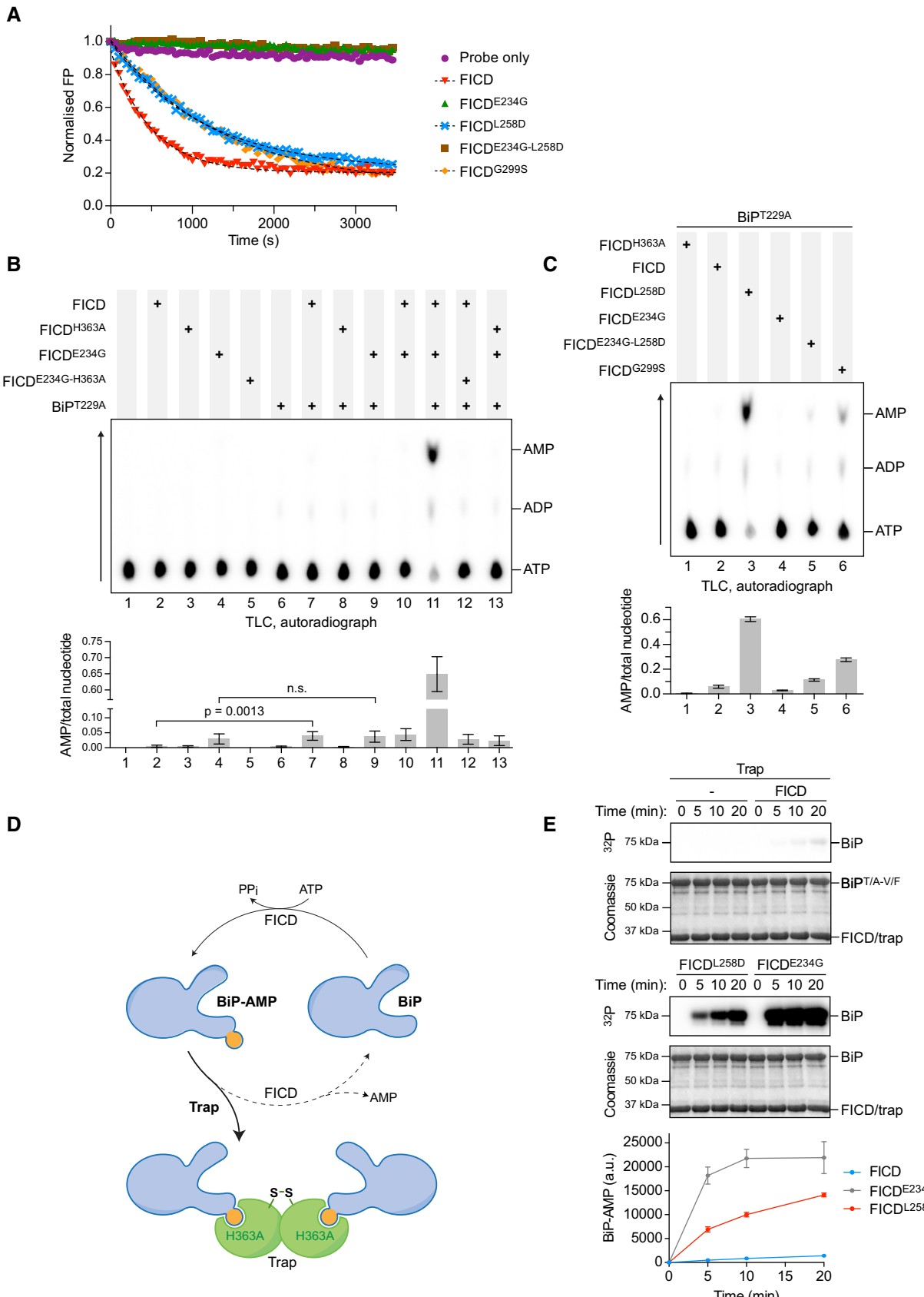

Figure 2.

**Figure 2. Monomerising mutations de-repress FICD's AMPylation activity.**

A  Monomerising FICD mutations inhibit deAMPylation. Shown is a representative plot of data points and fit curves of the time-dependent deAMPylation of a fluorescent BiP$^{V461F}$-AMP$^{FAM}$ by the indicated FICD proteins (at 7.5 μM) as detected by a change in fluorescence polarisation (FP). DeAMPylation rates calculated from independent experiments are given in Fig EV2A.

B, C  Dimer interface mutants both AMPylate and deAMPylate BiP. Shown are representative autoradiographs of thin-layer chromatography (TLC) plates revealing AMP produced from reactions containing [α-$^{32}$P]-ATP and the indicated FICD enzymes in the presence or absence of the co-substrate BiP (arrow indicates direction of nucleotide migration). The radioactive signals were quantified and the AMP signals were normalised to the total nucleotide signal in each sample. Plotted below are mean values ± SD from at least three independent experiments. Unpaired *t*-tests were performed. See Fig EV2D.

D  Cartoon depicting sequestration of AMPylated BiP by a covalently linked, disulphide-stapled, $_{S-S}$FICD$^{A252C-H363A-C421S}$ dimer (trap). See Fig EV2E–H.

E  Detection of the time-dependent accumulation of AMPylated BiP$^{T229A-V461F}$ in radioactive reactions, containing [α-$^{32}$P]-ATP and the indicated FICD proteins, in the presence of excess trap. At the specified time-points, samples were taken and analysed by SDS–PAGE. The autoradiograph ($^{32}$P) illustrates the radioactive signals, which represent AMPylated BiP. Proteins were visualised by Coomassie staining. The radioactive signals were quantified and presented in the graph below. Mean values ± SD of three independent experiments are shown.

Source data are available online for this figure.

disulphide-linked wild-type FICD ($_{S-S}$FICD$^{A252C-C421S}$), which, after purification and oxidation, formed a covalent dimer (Appendix Fig S1A). Moreover, its SEC profile was indistinguishable from wild-type FICD or the cysteine-free counterpart, FICD$^{C421S}$ (Appendix Fig S1B). In the presence of the BiP-AMP trap, oxidised $_{S-S}$FICD$^{A252C-C421S}$ produced significantly less AMPylated BiP than either wild-type or FICD$^{C421S}$ at similar concentrations (Fig 3B, lane 8 and Appendix Fig S1C).

Repression of AMPylation was imposed specifically by the covalent dimer, as non-oxidised FICD$^{A252C-C421S}$ elicited conspicuously more BiP-AMP than the wild-type enzyme (Fig 3B, lane 9 and Appendix Fig S1C)—an observation explained by the weakening of the FICD dimer imposed by the Ala252Cys mutation (Fig EV1D and E). Similarly, in absence of the trap, the ability of pre-oxidised $_{S-S}$FICD$^{A252C-C421S}$ to establish a pool of AMPylated BiP was greatly enhanced by diluting the enzyme into a buffer containing DTT. FICD$^{C421S}$, by contrast, produced similar amounts of modified BiP under both non-reducing and reducing conditions (Fig 3C).

DeAMPylation activities of oxidised and non-oxidised FICD$^{A252C-C421S}$ were comparable and similar to wild-type FICD (Figs 3D and E, and EV2A, and Appendix Fig S1D), pointing to the integrity of these mutant enzymes. Together, these observations argue that covalent $_{S-S}$FICD$^{A252C-C421S}$ dimers selectively report on the enzymatic characteristics of wild-type FICD in its dimeric state. This protein therefore serves to help validate the conclusion that a low concentration of wild-type FICD favours formation of monomers, whose AMPylation activity is de-repressed, and promotes BiP modification.

## An AMPylation-repressive signal is transmitted from the dimer interface to the active site

The crystal structure of dimeric FICD suggests the existence of a hydrogen-bond network, involving the side chains of Lys256 and Glu242, linking the dimer interface with the enzyme's active site, impinging on the AMPylation-inhibiting Glu234 (Fig 4A). To test this notion, we mutated both putative dimer relay residues. FICD$^{K256S}$ and FICD$^{E242A}$ formed stable dimers, as assayed by SEC, with dimer $K_d$ values under 400 nM (Figs 4B, and EV1D and E). *In vitro*, both mutants established a pool of modified BiP (Figs 4C and EV3A). This remained the case even at FICD concentrations in which negligible amounts of monomer are predicted (2 and 10 μM; Fig EV3A). De-repression of AMPylation by these dimer relay mutations was also evidenced by enhanced BiP-dependent AMP production,

relative to wild-type FICD (Fig 4D), whilst possessing similar deAMPylation activities (Figs EV2A and EV3B). Combining the Lys256Ser and the monomerising Leu258Asp mutations (FICD$^{K256S-L258D}$) further enhanced the BiP-AMP pool produced *in vitro* (Fig EV3A), an observation only partially attributable to the concomitant decrease in deAMPylation rate (Figs EV2A and EV3C). These observations suggest that residues connecting the dimer interface and the active site contribute to repression of AMPylation and that mutating these residues uncouples a gain-of-AMPylation activity from the oligomeric state of FICD.

Transmission of a repressive signal via a network of intramolecular interactions is also supported by the correlation between de-repression of BiP AMPylation and the negative effect of various mutants on the global stability of FICD. Differential scanning fluorimetry (DSF) revealed an inverse relationship between the AMPylation activity and the melting temperature ($T_m$) of FICD mutants (Figs 4E and EV3D). These differences in flexibility were observed despite the fact that the DSF assays were conducted at relatively high protein concentrations (2 μM) that would favour dimerisation of all but the most dimerisation-defective mutants.

Nucleotide binding stabilises all FICD variants (Fig EV3D), a feature that is conspicuous in the case of the AMPylation de-repressed FICD$^{E234G}$ (Bunney *et al*, 2014). However, monomerisation imposed by the Leu258Asp mutation did not significantly increase ATP-induced stabilisation of FICD ($\Delta T_m$; Figs 4F and EV3E). Interestingly, although AMPylation activity correlated with increased FICD flexibility, this was not reflected in an appreciably altered propensity to bind ATP. This suggested that the variation in enzyme activity of different FICD mutants may arise not from variation in their affinity for nucleotide but from their particular mode of ATP binding. To explore this possibility, we set out to co-crystallise FICD variants with MgATP.

## Monomerisation favours AMPylation-competent binding of MgATP

High-resolution X-ray crystal structures of monomeric and dimeric FICD were obtained in various nucleotide-bound states (Table 1 and Table EV1). The tertiary structure of the Fic domain of both the monomeric FICD$^{L258D}$ and the dimeric relay mutant FICD$^{K256S}$ deviated little from that of the nucleotide-free wild-type dimer structure (FICD:Apo; PDB: 4U04) (Figs 5A and EV4A). Moreover, co-crystallisation of FICD$^{L258D}$, FICD$^{K256A}$ or the wild-type dimer with ATP or

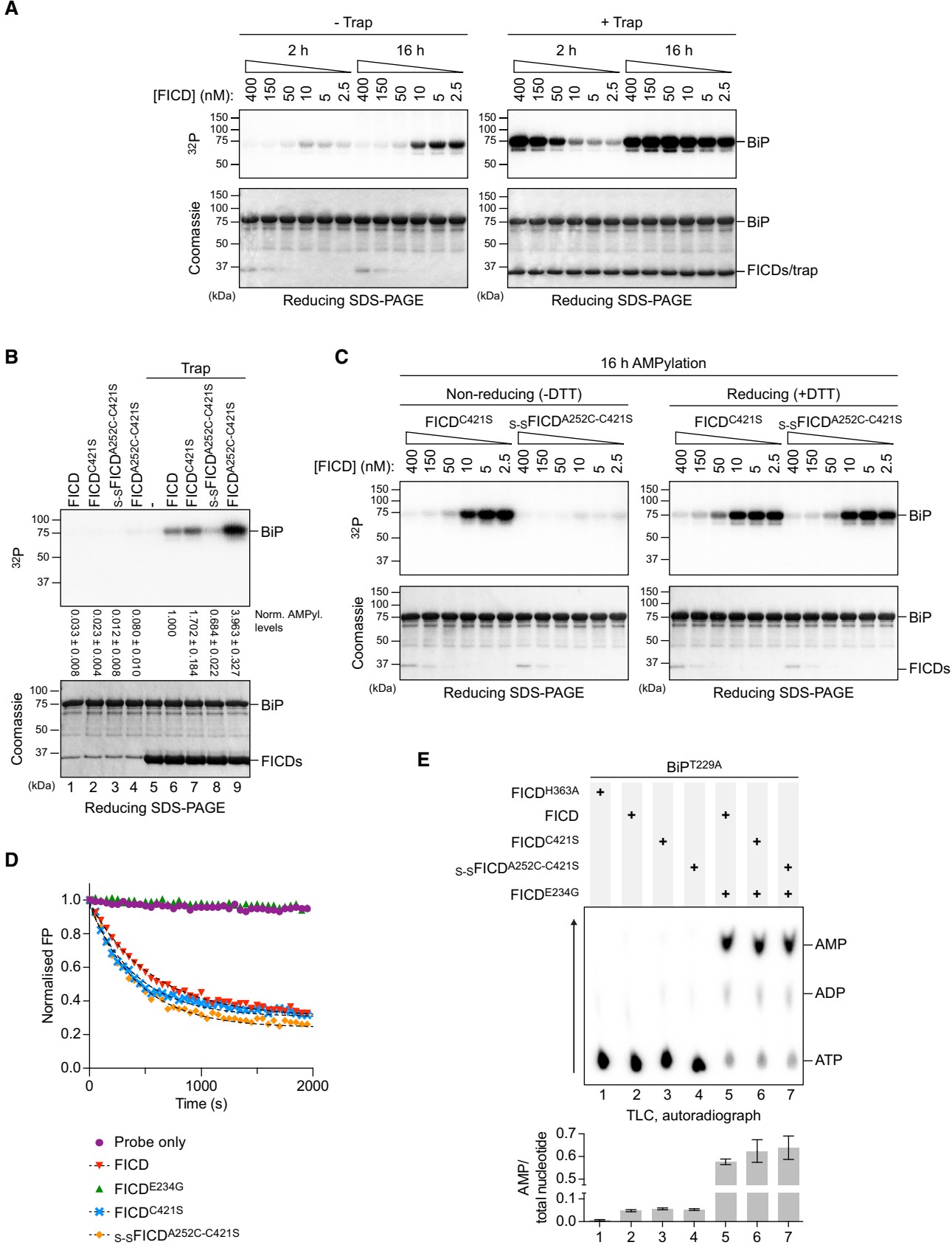

**Figure 3.**

◀

**Figure 3.** Monomerisation by dilution enhances the AMPylation activity of wild-type FICD.

A  Autoradiographs of *in vitro* reactions containing varying concentration of wild-type FICD protein and fixed concentrations of BiP$^{T229A-V461F}$ and [α-$^{32}$P]-ATP as co-substrates, resolved by SDS–PAGE after the indicated incubation times. The proteins were visualised by Coomassie staining of the gel (bottom). The reactions shown on the right were performed in the presence of an excess of $_{S-S}$FICD$^{A252C-H363A-C421S}$ (trap) to delay de-modification of BiP. Representative gels are shown, and similar results were observed in three independent experiments.

B  As in (A) but with 0.2 μM of the indicated FICD variant. The radioactive signals were detected by autoradiography, quantified and normalised to the signal in lane 6. The mean radioactive signals ± SD from three independent experiments are given. The proteins were visualised by staining with Coomassie. See Appendix Fig S1A and B.

C  As in (A) but with dilutions of FICD$^{C421S}$ or covalently linked $_{S-S}$FICD$^{A252C-C421S}$. Reactions were preceded by a 16-h incubation of FICD in presence or absence of the reducing agent (DTT). Representative gels are shown of three independent experiments. See Appendix Fig S1C.

D  Forced dimerisation does not significantly alter deAMPylation rates. Time-dependent deAMPylation of fluorescent BiP$^{V461F}$-AMP$^{FAM}$ by the indicated FICD proteins (at 7.5 μM) assayed by fluorescence polarisation (as in Fig 2A). A representative experiment (data points and fit curves) is shown, and rates are given in Fig EV2A. See Appendix Fig S1D.

E  Representative autoradiograph of thin-layer chromatography (TLC) plates revealing AMP produced from reactions containing [α-$^{32}$P]-ATP and the indicated FICD enzymes in the presence of the co-substrate BiP. AMP signals were normalised to the total nucleotide signal in each sample, and the graph below plots mean values ± SD from at least three independent experiments.

Source data are available online for this figure.

an ATP analogue (AMPPNP) also resulted in no significant Fic domain conformational change from FICD:Apo (Figs 5A and EV4A). Accordingly, the greatest root-mean-squared deviation (RMSD) between the Fic domain of the FICD:ATP structure and any other monomeric or dimer relay FICD structure is 0.53 Å (observed between FICD:ATP and FICD$^{L258D}$:Apo; residues 213–407). The only conspicuous change in global tertiary structure occurred in the TPR domain of FICD$^{L258D}$ co-crystallised with ATP or AMPPNP, in which the TPR domain is flipped almost 180° from its position in other FICD structures (Fig 5A). Notably, in all FICD structures the α$_{inh}$ remains firmly juxtaposed to the core Fic domain.

When co-crystallised with MgATP or MgAMPPNP, the resulting FICD structures contained clear densities for nucleotide (Figs 5B and EV4B). The AMPylation-biased FICD mutants also contained discernible, octahedrally coordinated Mg$^{2+}$ ions (Figs 5B(ii)–(iii) and EV4B). As noted in other Fic AMPylases, this Mg$^{2+}$ was coordinated by the α- and β-phosphates of ATP/AMPPNP and Asp367 of the Fic motif (Xiao *et al*, 2010; Bunney *et al*, 2014; Khater & Mohanty, 2015b). Interestingly, in the dimeric wild-type FICD:ATP structure, crystallised in the presence of MgATP, there was no density that could be attributed to Mg$^{2+}$ (Fig 5B(i)). The only possible candidate for Mg$^{2+}$ in this structure was a water density, located between all three phosphates, that fell in the Fic motif's anion hole —a position incompatible with Mg$^{2+}$ coordination (Zheng *et al*, 2017).

Alignment of the nucleotide-bound structures revealed that ATP or AMPPNP was bound very differently by the wild-type dimer and the AMPylation-biased monomeric or dimer relay FICD mutants (Figs 5C and EV4C). Concordantly, the RMSD of ATP between the wild-type FICD and monomeric FICD$^{L258D}$ was 2.17 Å (and 2.23 Å for FICD$^{K256A}$'s ATP). As previously observed in other ATP-bound Fic proteins that possess an inhibitory glutamate, the nucleotide in FICD:ATP was in an AMPylation non-competent conformation (Engel *et al*, 2012; Goepfert *et al*, 2013) that is unable to coordinate Mg$^{2+}$: an essential cation for FICD-mediated AMPylation (Ham *et al*, 2014). Moreover, the position of the ATP α-phosphate precludes in-line nucleophilic attack (by the hydroxyl group of BiP's Thr518) due to the proximity of the Fic domain flap residue Val316 (Figs 5C and EV4D). Furthermore, an attacking nucleophile in-line with Pα-O3α would be at a considerable distance from the catalytic

His363 (required to deprotonate Thr518's hydroxyl group; Figs 5B(i) and C, and EV4D).

By contrast, in the active sites of FICD$^{K256A}$ or FICD$^{L258D}$ MgATP and MgAMPPNP assumed AMPylation-competent conformations: their α-phosphates were in the canonical position (Fig EV4E), as defined by AMPylation-active Fic proteins lacking inhibitory glutamates (Xiao *et al*, 2010; Engel *et al*, 2012; Goepfert *et al*, 2013; Bunney *et al*, 2014). As a result, in-line nucleophilic attack into the α-β-phosphoanhydride bond of ATP would not be sterically hindered and the Nε2 of His363 would be well positioned for general base catalysis (Figs 5C, and EV4C and D).

The presence of ATP in both dimeric wild-type FICD and monomeric FICD$^{L258D}$ (although in different binding modes) is consonant with the DSF data (Figs 4F and EV3E). Apart from Glu234, the residues directly interacting with ATP are similarly positioned in all structures (maximum RMSD 0.83 Å). However, considerable variability is observed in Glu234, with an RMSD of 4.20 Å between monomeric and dimeric wild-type ATP structures, which may hint at the basis of monomerisation-induced AMPylation competency. In all ATP-bound structures, the inhibitory glutamate is displaced from the respective apo ground-state position, in which it forms an inhibitory salt-bridge with Arg374 (R$_2$ of the Fic motif; Appendix Fig S2A). However, the displacement of the Glu234 side chain observed in the FICD:ATP structure (from its position in FICD:Apo; PDB 4U04) would be insufficient for AMPylation-competent binding of the γ-phosphate of an ATP/AMPPNP (see distances (i) and (ii), Figs 5C and EV4C). This steric clash is relieved by the side chain conformations observed in the AMPylation-competent structures (see (iii) and (iv), Figs 5C and EV4C).

The findings above suggest that the AMPylation-biased FICD mutants attain their ability to competently bind MgATP by increased flexibility at the top of the α$_{inh}$ and by extension through increased Glu234 dynamism. It is notable that all the nucleotide triphosphate-bound FICDs crystallised with intact dimer interfaces (Appendix Fig S2A and B). Moreover, with the exception of direct hydrogen bonds to mutated Lys256 side chains, in all FICD crystals the putative dimer relay hydrogen-bond network was maintained (Appendix Fig S2A). It seems likely that much of the monomerisation-linked conformational flexibility that facilitates binding of MgATP in

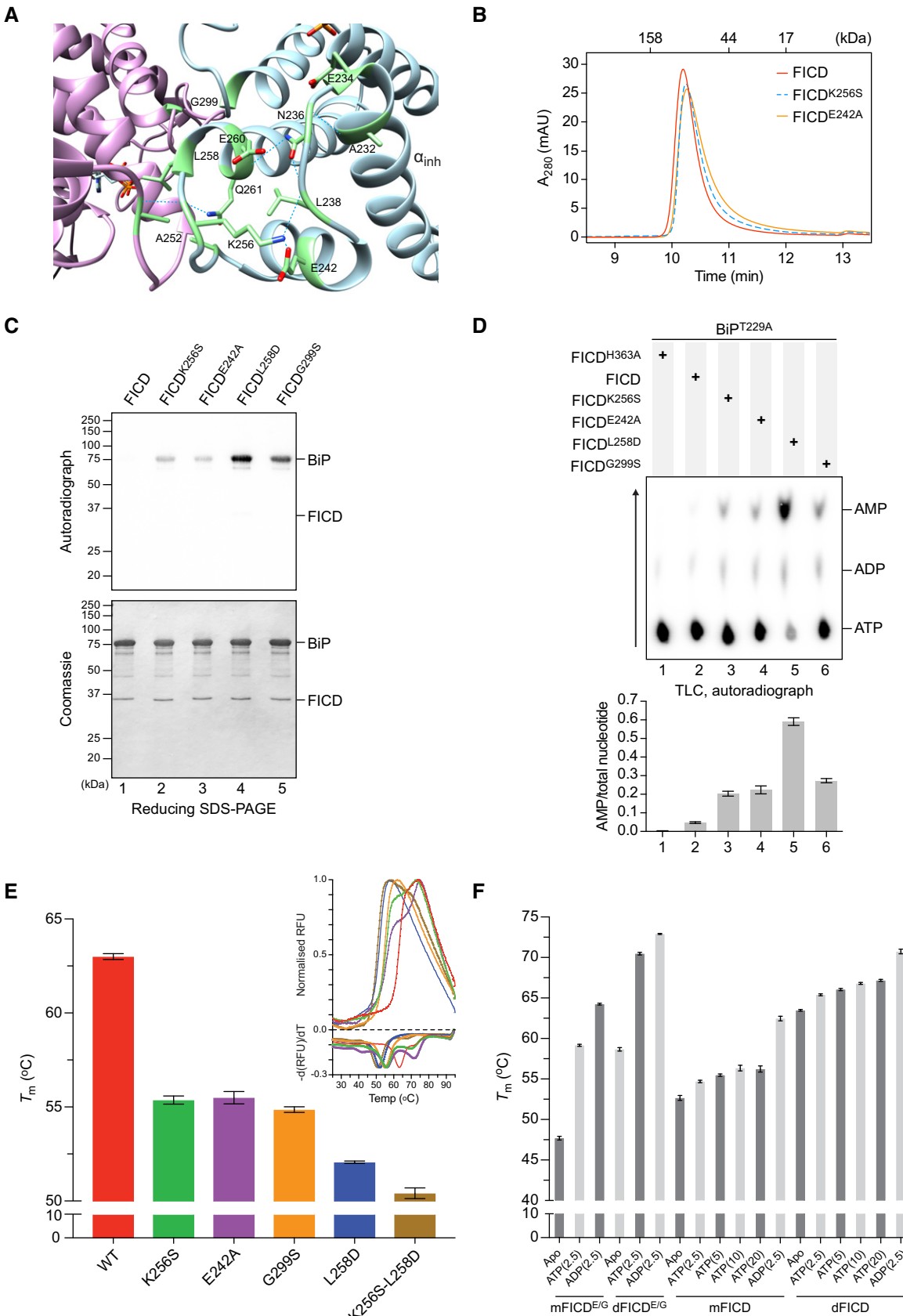

**Figure 4.**

Figure 4. Residues connecting the FICD dimer interface with the inhibitory α-helix stabilise FICD and repress AMPylation.

A  Ribbon diagram of the FICD dimer interface with monomers in purple and blue ribbons (PDB: 6I7G). Residues involved in a H-bond network linking the dimer interface to the $\alpha_{inh}$ (as well as Gly299 and Glu234) are shown as green sticks. Sub-3.50 Å hydrogen bonds made by Asn236, Leu238 and Lys256 are depicted as dotted cyan lines.

B  Size-exclusion chromatography (SEC) elution profile of wild-type and mutant FICD proteins (each at 20 μM). Protein absorbance at 280 nm is plotted against elution time. The elution times of protein standards are indicated as a reference. mAU, milli-absorbance units.

C  Radioactive in vitro AMPylation reactions containing the indicated FICD proteins, $[\alpha-^{32}P]$-ATP and BiP$^{T229A-V461F}$ were analysed by SDS–PAGE. The radioactive BiP-AMP signals were detected by autoradiography, and proteins were visualised by Coomassie staining of the gel. See Fig EV3A.

D  Representative autoradiograph of thin-layer chromatography (TLC) plates revealing AMP produced from reactions containing $[\alpha-^{32}P]$-ATP and the indicated FICD enzymes in the presence of the co-substrate BiP. The radioactive signals were quantified and the AMP signals were normalised to the total nucleotide signal in each sample. The graph shows mean AMP values ± SD from three independent experiments.

E  Melting temperatures ($T_m$) of the indicated FICD mutants (at 2 μM) were measured by differential scanning fluorimetry (DSF). Shown is the mean $T_m$ ± SD of three independent experiments. The inset shows melt curves with their negative first derivatives from a representative experiment. See Fig EV3D. RFU, relative fluorescence units.

F  A plot of the melting temperature of the indicated FICD proteins in absence (Apo) or presence of nucleotides. Shown are the mean $T_m$ values ± SD of three independent DSF experiments. Monomeric FICD$^{L258D}$ (mFICD) and FICD$^{L258D-E234G}$ (mFICD$^{E/G}$) as well as dimeric wild-type FICD (dFICD) and FICD$^{E234G}$ (dFICD$^{E/G}$) were tested. ADP and ATP concentrations in mM are given in parentheses. See Fig EV3E for $K_{1/2}$ quantification.

Source data are available online for this figure.

## Table 1. Data collection and refinement statistics.

| | FICD:ATP | FICD$^{K256S}$:Apo | FICD$^{K256A}$:MgATP | FICD$^{L258D}$:Apo | FICD$^{L258D}$:MgATP | FICD$^{L258D}$:MgAMPPNP |
|---|---|---|---|---|---|---|
| **Data collection** | | | | | | |
| Synchrotron stations | DLS I04 | DLS I03 | DLS I03 | DLS I03 | DLS I03 | DLS I03 |
| Space group | $P2_12_12$ | $P22_12_1$ | $P22_12_1$ | $P3_121$ | $P6_422$ | $P6_422$ |
| Molecules in a.u.$^a$ | 2 (2) | 1 (2) | 1 (2) | 1 (1) | 1 (1) | 1 (1) |
| a,b,c; Å | 77.67, 107.65, 132.60 | 43.82, 76.51, 131.97 | 41.90, 73.98, 134.04 | 118.14, 118.14, 79.55 | 186.84, 186.84, 76.84 | 186.36, 186.36, 77.10 |
| α, β, γ; ° | 90.00, 90.00, 90.00 | 90.00, 90.00, 90.00 | 90.00, 90.00, 90.00 | 90.00, 90.00, 120.00 | 90.00, 90.00, 120.00 | 90.00, 90.00, 120.00 |
| Resolution, Å | 83.58–2.70 (2.83–2.70) | 65.99–2.25 (2.32–2.25) | 134.04–2.32 (2.41–2.32) | 62.80–2.65 (2.72–2.65) | 93.42–2.54 (2.65–2.54) | 93.18–2.31 (2.39–2.31) |
| $R_{merge}$ | 0.163 (0.717) | 0.109 (0.385) | 0.107 (0.636) | 0.176 (0.856) | 0.167 (1.009) | 0.071 (0.611) |
| $<I/\sigma(I)>$ | 19.2 (1.8) | 6.8 (2.4) | 5.6 (1.0) | 8.6 (2.2) | 13.0 (2.5) | 10.3 (1.8) |
| CC1/2 | 0.999 (0.720) | 0.993 (0.547) | 0.995 (0.567) | 0.996 (0.549) | 0.999 (0.503) | 0.998 (0.523) |
| No. of unique reflections | 31,293 (4,091) | 21,825 (1,978) | 18,543 (1,712) | 18,963 (1,380) | 26,617 (3,188) | 34,573 (3,351) |
| Completeness, % | 100.0 (100.0) | 99.9 (99.5) | 99.4 (97.3) | 100.0 (100.0) | 100.0 (100.0) | 99.4 (99.1) |
| Redundancy | 6.4 (6.5) | 4.4 (4.4) | 3.7 (3.7) | 9.7 (10.0) | 16.1 (16.5) | 4.6 (4.6) |
| **Refinement** | | | | | | |
| $R_{work}/R_{free}$ | 0.280/0.319 | 0.208/0.259 | 0.282/0.325 | 0.228/0.283 | 0.232/0.252 | 0.214/0.251 |
| No. of atoms (non-H) | 5,650 | 2,851 | 2,731 | 2,951 | 2,828 | 2,940 |
| Average B-factors, Å$^2$ | 55.3 | 42.5 | 54.6 | 50.9 | 58.2 | 56.4 |
| RMS Bond lengths, Å | 0.002 | 0.003 | 0.003 | 0.003 | 0.002 | 0.003 |
| RMS Bond angles, ° | 1.142 | 1.180 | 0.763 | 1.222 | 1.127 | 1.170 |
| Ramachandran favoured region, % | 96.5 | 98.5 | 98.2 | 97.9 | 98.5 | 99.4 |
| Ramachandran outliers, % | 0 | 0 | 0 | 0 | 0 | 0 |
| MolProbity score$^b$ | 1.33 (100$^{th}$) | 0.86 (100$^{th}$) | 0.74 (100$^{th}$) | 0.99 (100$^{th}$) | 0.97 (100$^{th}$) | 0.99 (100$^{th}$) |
| PDB code | 6I7G | 6I7H | 6I7I | 6I7J | 6I7K | 6I7L |

Values in parentheses correspond to the highest-resolution shell, with the following exceptions: $^a$The number of molecules in the biological unit is shown in parentheses; $^b$MolProbity percentile score is shown in parentheses (100$^{th}$ percentile is the best among structures of comparable resolutions; 0$^{th}$ percentile is the worst). For crystallisation conditions see Table EV1.

solution cannot be observed crystallographically. Nonetheless, comparing B-factors across the nucleotide triphosphate-bound FICD structures is informative: despite similar crystal packing

(Appendix Fig S2B), the average residue B-factors, both in the dimerisation interface and near Glu234, positively correlated with the AMPylation activities of the respective mutants (Fig EV5).

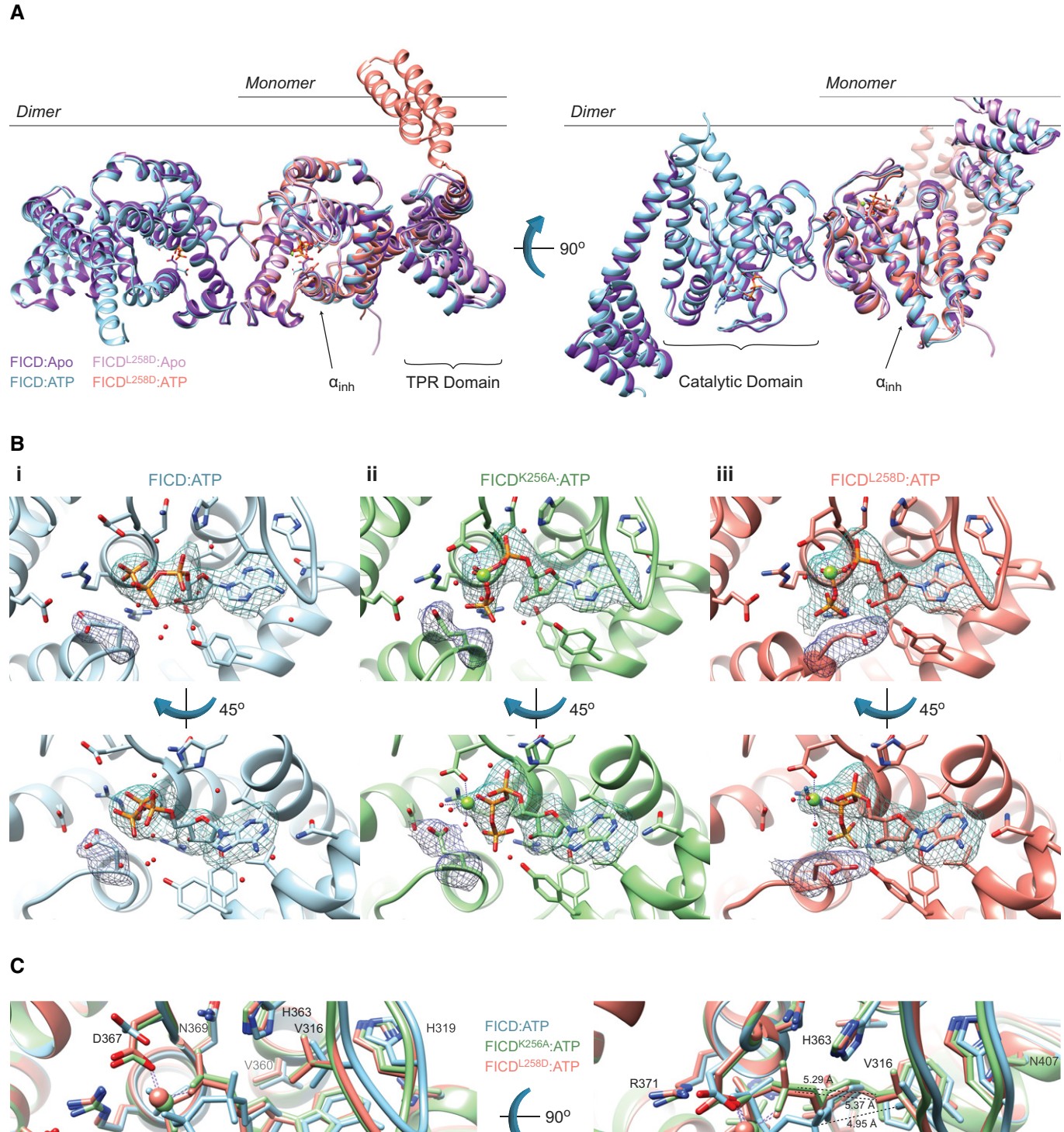

**Figure 5.**

**Figure 5. Monomeric FICD binds ATP in an AMPylation-competent conformation.**

A  Monomerisation does not result in large conformational changes in FICD. Shown is the alignment, from residues 213–407, of FICD molecules in the asymmetric unit. Monomeric FICD$^{L258D}$ and dimeric wild-type FICD ± ATP are coloured as indicated. Glu234, ATP (and Mg, where applicable), is shown as sticks (or green spheres). The inhibitory alpha helix ($\alpha_{inh}$) and gross domain architecture are annotated. Note the only significant deviation in tertiary structure is the flipping of the TPR domain in the FICD$^{L258D}$:ATP structure. The FICD:Apo structure is from PDB: 4U04. See Fig EV4A.

B  Co-crystallisation of FICD variants with MgATP results in electron densities for nucleotide and the inhibitory Glu234. Unbiased polder (OMIT) maps for ATP (± Mg) and Glu234 are shown as blue and purple meshes, respectively. (i) The wild-type dimer FICD structure displays a lack of density corresponding to a Mg$^{2+}$ ion. The ATP density is contoured at 3.5σ and the Glu234 at 5.0σ. (ii) The dimeric dimer relay mutant FICD$^{K256A}$ displays a clear MgATP density up to and including the γ-phosphate phosphorous atom. The ATP density and Glu234 densities are both contoured at 3.0σ. (iii) Monomeric FICD$^{L258D}$ shows a clear MgATP density. The ATP density is contoured at 3.0σ and the Glu234 at 5.0σ. All residues and water molecules interacting with ATP (± Mg) are shown as sticks and coloured by heteroatom. Mg$^{2+}$ coordination complex pseudo-bonds are shown in purple dashed lines. See Fig EV4B.

C  Unlike the monomeric or the dimer relay FICD mutants, dimeric wild-type FICD binds ATP in a configuration that would prevent BiP substrate AMPylation. The position of the α-phosphate in the FICD:ATP structure would preclude in-line nucleophilic attack (see Fig EV4C and D). The left panel represents the superposition of the structures in the upper panel of (B), with ATP interacting residues shown as sticks and annotated. Only Glu234 deviates significantly in side chain position. Note, however, that the FICD:ATP His363 side chain is also flipped, forming a hydrogen bond to a ribose interacting water (see B(i)). Mg$^{2+}$ and ATP are coloured to match the corresponding ribbons. Active site waters are omitted for clarity. Distances are indicated by dashed black lines. The inset is a blow-up displaying distances (i–iv) between the γ-phosphates and Glu234 residues. Note, distances (i) and (ii) are derived from the γ-phosphate and Glu234 of different superimposed structures. Distances between Val316(Cγ1) and the corresponding Pα are shown in the right-hand side panel.

## ATP is an allosteric modulator of FICD

Given the conspicuous difference in the ATP binding modes observed between AMPylation-competent FICD mutants and the AMPylation-incompetent wild-type dimeric FICD, we were intrigued by the possibility that ATP may modulate other aspects of FICD enzymology and regulation.

In order to explore the effects of nucleotide on the different pre-AMPylation complexes formed between either dimeric or monomeric FICD and its co-substrate, ATP-bound BiP, we utilised BioLayer interferometry (BLI). Biotinylated, client-binding-impaired, ATPase-defective BiP$^{T229A-V461F}$ was made nucleotide-free (apo) and immobilised on a streptavidin biosensor. Its interactions with catalytically inactive, dimeric FICD$^{H363A}$ or catalytically inactive, monomeric FICD$^{L258D-H363A}$ were measured in the presence and absence of nucleotides. The binding of both monomeric and dimeric FICD to immobilised BiP was greatly enhanced by the pre-saturation of BiP with ATP (Figs 6A and EV6A). This is consistent with ATP-bound BiP serving as the substrate for FICD-mediated AMPylation (Preissler *et al*, 2015b). Moreover, the binding signal from the interaction of immobilised ATP-bound BiP with monomeric FICD$^{L258D-H363A}$:Apo was significantly stronger than that produced from the corresponding dimeric FICD$^{H363A}$:Apo analyte (Fig 6A). In contrast, with respect to pre-deAMPylation complex assembly, AMPylated BiP bound more tightly to dimeric FICD$^{H363A}$ than to monomeric FICD$^{L258D-H363A}$ (Fig EV2G). These findings align with a role of dimeric FICD in deAMPylation and the monomer in AMPylation.

Interestingly, in the presence of magnesium and nucleotide (either MgATP or MgADP), the FICD$^{H363A}$ interaction with ATP-bound BiP was weakened (Fig 6A). This effect was considerably more pronounced for monomeric FICD$^{L258D-H363A}$. To quantify the effect of FICD monomerisation on the kinetics of pre-AMPylation complex dissociation, BLI probes pre-assembled with biotinylated, ATP-bound BiP and either apo dimeric FICD$^{H363A}$ or apo monomeric FICD$^{L258D-H363A}$ were transferred into otherwise identical solutions ± ATP (schematised in Fig EV6B). The ensuing dissociations fit biphasic exponential decays and revealed that ATP binding to FICD accelerated the dissociation of monomeric FICD$^{H363A}$ more than dimeric FICD$^{H363A}$ (Figs 6B and EV6C). The effect of ATP was noted on both the rate of the slow dissociation phase ($k_{off,slow}$;

Fig 6C and D) and on the percentage of the dissociation attributed to the fast phase (%Fast; Figs 6D and EV6D). The effect of ATP on the dissociation kinetics of the FICD$^{L258D-H363A}$/BiP:ATP complex, measured under conditions of effectively infinite dilution, argues against a simple one-site competition between ATP-bound BiP and ATP for the Fic domain active site. Instead, these observations are better explained by ATP allosterically modulating monomeric FICD.

The structural data revealed that FICD's oligomeric state can impact significantly on the mode of ATP binding, and Fig 6B indicated that there is an allosteric effect of nucleotide binding on FICD. Together, these observations suggested bidirectional intramolecular signalling from the dimer interface to the nucleotide-binding active site and the possibility that ATP binding in FICD's active site may also influence the oligomeric state of the protein. To investigate this hypothesis, hetero-dimers of N-terminally biotinylated FICD$^{H363A}$ assembled with non-biotinylated FICD$^{H363A}$ were loaded onto a BLI streptavidin biosensor. The dissociation of non-biotinylated FICD$^{H363A}$ from its immobilised partner was then observed by quasi-infinite dilution into buffers varying in their nucleotide composition (Figs 6E, and EV6E and F). ATP but not ADP induced a 3-fold increase in the dimer dissociation rate (Fig 6E). This is suggestive of a mechanism whereby changing ATP/ADP ratios in the ER may modulate the oligomeric state of FICD but does not preclude the possibility of a compensatory increase in dimer association rate.

To directly assess the effect of nucleotide on the FICD monomer-dimer equilibrium, we developed a dimerisation-sensitive FICD fluorescent probe. Site-specific tetramethylrhodamine (TMR)-labelling of Ser288Cys (in a catalytically inactive and otherwise cysteine-free FICD$^{H363A-C421S}$ background) facilitated an optical readout of FICD's oligomeric state. The inter-fluorophore distance permitted by FICD dimerisation was predicted to attenuate the TMR fluorescence signal, as compared to the FICD-TMR monomer. Indeed, incubation of the probe with an escalating concentration of unlabelled dimerisation-competent FICD, but not unlabelled monomeric FICD$^{L258D}$, led to a progressive increase in the fluorescence of FICD-TMR (Fig 6F). This observation is well explained by a reduction in fluorophore self-quenching as FICD-TMR dimers are converted to hetero-dimers containing only one labelled protomer.

This dimerisation-sensitive probe was used to investigate the ability of ATP and ADP to modulate the FICD monomer-dimer

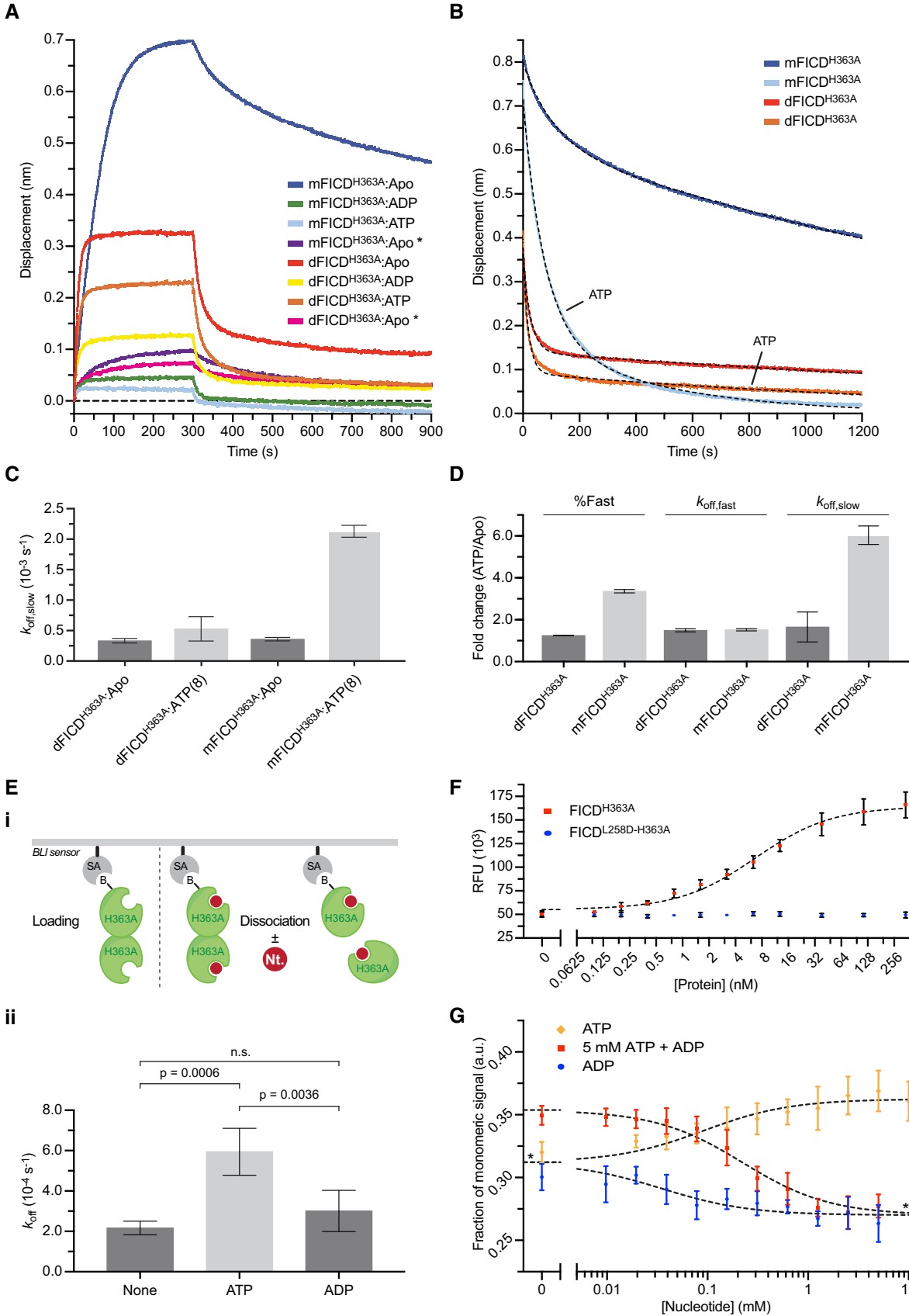

**Figure 6.**

◀

**Figure 6. ATP destabilises the pre-AMPylation complex and the FICD dimer.**

A  BioLayer interferometry (BLI) derived association and dissociation traces of monomeric FICD[L258D-H363A] (mFICD[H363A]) or dimeric FICD[H363A] (dFICD[H363A]) from immobilised biotinylated BiP[T229A-V461F] in absence or presence of nucleotides. Unless indicated (*), BiP was saturated with ATP before exposure to FICD variants. A representative experiment of three independent repetitions is shown. See Fig EV6A and B.

B  BLI dissociation traces of proteins as in (A). At $t = 0$ a pre-assembled complex of immobilised, ATP-saturated BiP and the indicated FICD proteins (associated without ATP) were transferred into a solution without or with ATP, as indicated. A representative experiment is shown and the biphasic dissociation kinetics are quantified in (C) and (D). Full association and dissociation traces are shown in Fig EV6C.

C  Graph of the slow dissociation rates ($k_{off,slow}$) of FICD from BiP:ATP, as derived from the data represented in (B). Bars represent mean values ± SD of three independent experiments.

D  The ATP-induced fold change in the percentage of the dissociation phase attributed to a fast dissociation (%Fast), $k_{off,fast}$, and $k_{off,slow}$, derived from the data represented in (B). Bars represent mean values ± SD of three independent experiments. See Fig EV6D.

E  (i) BLI workflow used to assay FICD dimer off rate data presented in (ii) and Fig EV6E and F. (ii) Observed dimer off rates under different nucleotide conditions (5 mM, where applicable). ATP, but not ADP, significantly increases the dimer dissociation rate (n.s.: not significant by Tukey test). Data shown are the mean ± SD of four independent experiments. See Fig EV6E and F.

F  Validation of the fluorescent dimerisation probe. The dimerisation-sensitive TMR fluorescence of the labelled dimer (2.5 nM) is de-quenched specifically by equilibration with excess unlabelled dimerisation-competent FICD[H363A] but not monomeric FICD[L258D-H363A]. Mean ± SD of three independent experiments. RFU, relative fluorescence units.

G  Fluorescence measurement of nucleotide-dependent modulation of the FICD monomer-dimer equilibrium. ATP increases and ADP decreases the proportion of monomeric FICD. *Plateaus were constrained to a shared best-fit value. Data shown are the mean ± SD of four independent experiments. a.u., arbitrary units.

Source data are available online for this figure.

equilibrium. In agreement with the BLI experiment that reported on ATP-enhanced FICD dimer dissociation rate, ATP was observed to increase the proportion of monomer in a concentration-dependent fashion (Fig 6G). ADP, on the other hand, pushed the FICD equilibrium towards the dimer and also effectively antagonised the monomerising effect of ATP (Fig 6G). Together, these observations attest to a coupling of FICD's oligomeric state to the identity of the bound nucleotide.

## Discussion

In order to match the folding capacity of the ER to the burden of unfolded proteins, independently of changes in gene expression, FICD catalyses BiP AMPylation and deAMPylation. Our study addresses a key process in which this single bifunctional enzyme switches between these two mutually antagonistic activities, thereby facilitating a post-translational UPR. The high affinity of FICD protomers for each other specifies the presence of principally dimeric FICD in the ER, shown here to restrict the enzyme to deAMPylation. This is the dominant mode of FICD both *in vitro* and in cells under basal conditions (Casey *et al*, 2017; Preissler *et al*, 2017a). However, establishing a pool of monomeric FICD unmasks its potential as a BiP AMPylase and enfeebles deAMPylation. The structural counterpart to this switch is the mode by which MgATP, the AMPylation reaction's co-substrate, is productively engaged in the active site of the enzyme. Our studies suggest that monomerisation relieves the repression imposed on FICD AMPylation by weakening a network of intramolecular contacts. In the repressed state, these contacts propagate from the dimer interface to the enzyme's active site and stabilise a conserved inhibitory residue, Glu234, which blocks AMPylation-competent binding of MgATP (Fig 7).

Our observations of a biphasic FICD concentration-dependent rescue of BiP AMPylation in *FICD*[−/−] cells, and the conspicuous ability of the monomerising Leu258Asp mutation to establish a modified BiP pool in *FICD*[−/−] cells, support an oligomeric state-dependent switch as a contributor to FICD regulation *in vivo*. This case is further reinforced by the divergent enzymatic properties of

monomeric mutants and enforced disulphide-linked dimers *in vitro*, and by measurements of the enzymatic activity of wild-type FICD in concentration regimes above and close to the dimerisation $K_d$. Complete monomerisation resulted in a 19-fold increase in AMPylation activity and a 2-fold decrease in deAMPylation activity. The concordance between monomeric FICD[L258D], dimerisation-defective mutants and mutants in the repressive relay from the dimer interface to the active site gives confidence in the validity of the biophysical and structural insights provided by the mutants.

The inverse correlation observed between the thermal stability of FICD mutants and their AMPylation activities supports a role for enhanced flexibility in enabling the enzyme to attain the conformation needed for catalysis of this reaction—a role clarified by the crystallographic findings (see below). The biophysical assays also suggest that monomeric FICD is more allosterically sensitive to ATP binding as it exhibits a pronounced nucleotide-dependent reduction in the affinity for its co-substrate, ATP-bound BiP. The observation that ATP significantly accelerated the dissociation of monomeric, nucleotide-free FICD from ATP-bound BiP suggests that this feature of the monomer is mediated allosterically (not by enhanced susceptibility of a destabilised protein to co-substrate competition for the same active site). The lower affinity of monomeric FICD for its BiP: ATP co-substrate, in the context of a quaternary pre-AMPylation complex, conspicuously distinguishes it from the dimer. We speculate that this feature may also enhance AMPylation rates, as ground-state destabilisation has been demonstrated in a number of enzymes to enhance catalytic rate by weakening the otherwise anti-catalytic tight binding of an enzyme to its substrate (Andrews *et al*, 2013; Ruben *et al*, 2013).

A structure of the quaternary pre-AMPylation complex, that could inform our understanding of the features of the monomeric enzyme, does not exist. Nevertheless, important insights into the effect of monomerisation were provided by crystal structures of FICD and its nucleotide co-substrate. Dimeric wild-type FICD binds ATP (without magnesium) in an AMPylation-incompetent mode. This is consistent with all other inhibitory glutamate-containing Fic structures crystallised with ATP or ATP analogues (Engel *et al*, 2012; Goepfert *et al*, 2013). In stark contrast, we have discovered

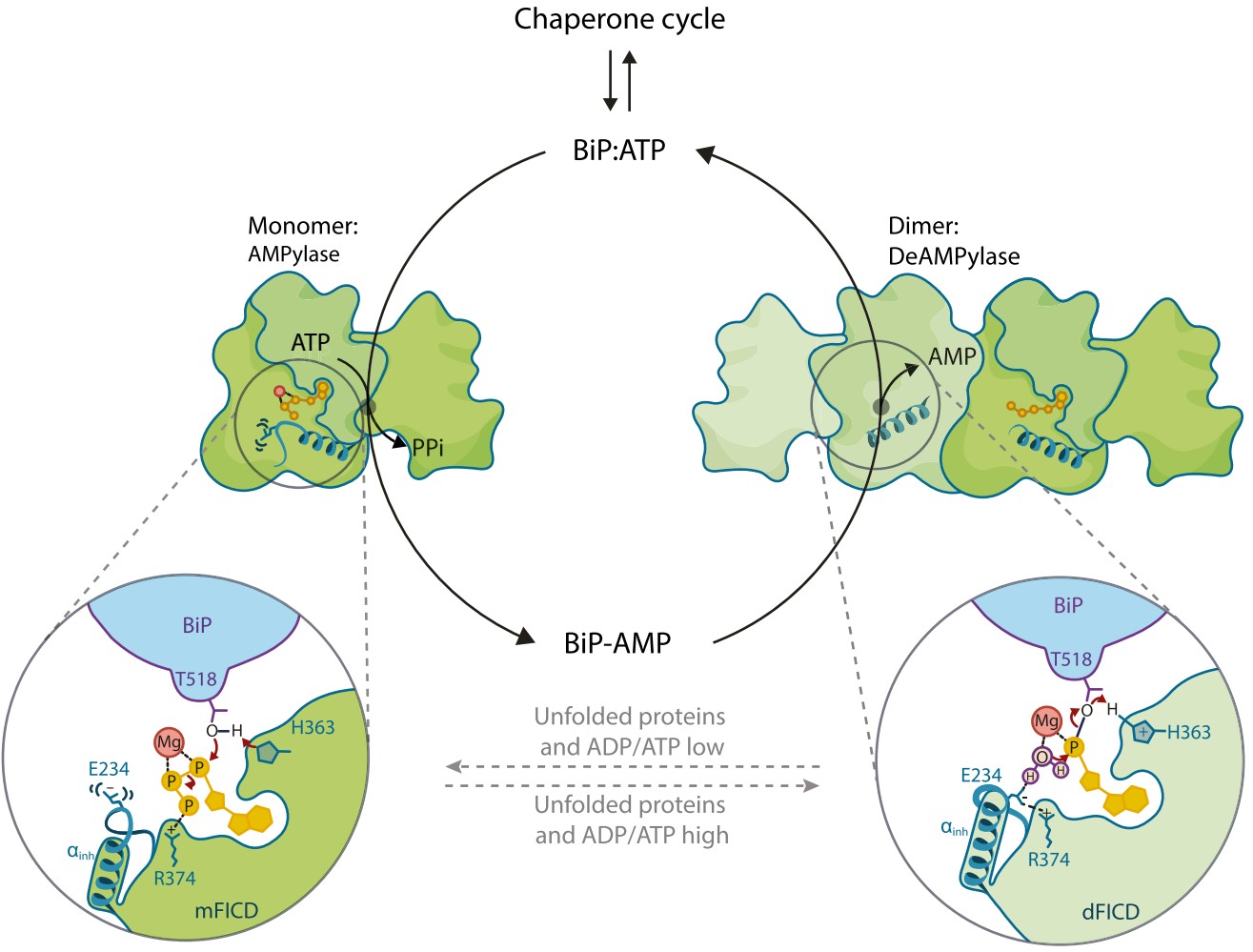

**Figure 7.   A proposed model of an oligomerisation state-dependent switch in FICD bifunctional active site.**

Under conditions of ER stress, the dimeric form of FICD is favoured (right-hand side). Dimeric FICD (dFICD) cannot bind ATP in an AMPylation-competent mode but can efficiently catalyse deAMPylation of BiP-AMP (thereby remobilising BiP back into the chaperone cycle). A decrease in unfolded protein load in the ER, possibly associated with a decreased ER ADP/ATP ratio, shifts the FICD monomer-dimer equilibrium towards monomeric FICD (mFICD). Monomeric FICD can bind MgATP in an AMPylation-competent conformation and, as such, AMPylate and inactivate surplus BiP.

that monomerisation, or mutations in residues linking the dimer interface to the active site, permits the binding of ATP with magnesium in a conformation competent for AMPylation despite the presence of the inhibitory Glu234.

The disparities in the manner of FICD-ATP binding are well explained by a monomerisation-induced increase in Glu234 flexibility (mediated by weakening of the dimer relay), as reflected in the lower melting temperatures of FICD[K256A] and FICD[L258D] relative to the wild-type dimer and localised B-factor increases in their respective crystal structures. In solution, monomerisation seems likely to allow even greater flexibility in the dimer relay network, facilitating motion and possibly unfolding at the top of the Glu234 containing α-helix ($\alpha_{inh}$). Such considerations could explain the comparatively small differences in the position of Glu234, but stark differences in nucleotide conformation, observed between the dimeric wild-type and monomeric or dimer relay mutant structures. That is to say, in solution the mutants exhibit sufficiently increased Glu234 dynamics

to permit binding of MgATP in a catalytically competent mode. However, the crystallisation process quite possibly favours rearrangements, including $\alpha_{inh}$ refolding and crystallographic reconstitution of the dimer interface, and convergence towards a low energy state (the one stabilised in solution by dimerisation). This then outweighs the energetic penalty of the resulting (crystallographically induced) electronically or sterically strained carboxylate-carboxylate (Glu234–Glu263) or glutamate-phosphate contacts (Figs 5C and EV4C). Crystallisation may therefore facilitate the apparent convergence of mutant FICD Glu234 conformations towards that imposed in solution by the dimer. By contrast, dimeric wild-type FICD is never able to bind MgATP competently, either in solution or *in crystallo*, due to its unperturbed allosteric dimer relay and consequently inflexible Glu234.

Oligomerisation state-mediated regulation of AMPylation is not unique to FICD. Tetramerisation of bacterial NmFic antagonises auto-AMPylation and AMPylation of its substrate, DNA gyrase

(Stanger *et al*, 2016). Though the surfaces involved in oligomerisation of this class III Fic protein are different from that of FICD, these two repressive mechanisms converge on the state of their $\alpha_{inh}$s. As such, divergent Fic proteins potentially exploit, for regulatory purposes, an intrinsic metastability of this structurally conserved inhibitory α-helix (Garcia-Pino *et al*, 2008). Interestingly, the more extensive dimerisation surface of FICD (which contains Leu258 and is situated at the boundary of the Fic domain core and the N-terminal Fic domain extension) also acts as a structurally conserved dimer interface in other class II bacterial Fic proteins: CdFic (Dedic *et al*, 2016) and *Bacteroides thetaiotaomicron* (BtFic; PDB: 3CUC), but not in the monomeric *Shewanella oneidensis* Fic (SoFic) protein (Goepfert *et al*, 2013). Moreover, a His57Ala mutation in dimeric CdFic (which is structurally equivalent to FICD$^{K256A}$) causes increased solvent accessibility and auto-AMPylation of a region homologous to the loop linking FICD's Glu242-helix and the $\alpha_{inh}$ (Dedic *et al*, 2016). Despite differences in detail, these findings suggest the conservation of a repressive relay from the dimer interface to the active site of dimeric Fic proteins.

The biophysical observations also suggest a reciprocal allosteric signal propagated from FICD's nucleotide-binding site back to the dimer interface. Enhanced dimer dissociation was induced by ATP but not ADP and an increasing ADP/ATP ratio was also observed to modulate the monomer-dimer equilibrium in favour of dimerisation *in vitro*. Consequently, it is tempting to speculate that FICD's oligomeric state and hence enzymatic activity might be regulated by the ADP/ATP ratio in the ER. Under basal conditions, low ADP concentrations allow ATP to bind both the monomeric and dimeric pools of FICD, shifting the equilibrium towards the monomer and favouring BiP AMPylation. Stress conditions may increase ADP concentration in the ER (perhaps as a consequence of enhanced ER chaperone ATPase activity). This increase would be proportionally much greater than the concomitant decrease in ATP concentration (in terms of respective fold changes in concentration). The increased ADP/ATP ratio would therefore result in a greater ADP fractional occupancy of FICD (and decreased ATP occupancy) and thereby shift the monomer-dimer equilibrium back towards the BiP de-AMPylating FICD dimer.

Additional layers of FICD regulation may exist alongside this hypothesised mechanism. A reduced load of unfolded protein may decrease ER-luminal molecular crowding and increase BiP AMPylation by lowering the effective concentration of FICD, promoting its monomerisation. Such excluded volume effects have been demonstrated for other homodimeric proteins *in vitro* (Wilf & Minton, 1981; Patel *et al*, 2002). Moreover, as BiP client concentration declines in the ER, the increase in ATP-bound (clientless) BiP may favour AMPylation simply by enhancing FICD substrate availability. This possibility is supported by the observation that cells overexpressing fully monomeric FICD$^{L258D}$ nonetheless exhibit an increase in BiP-AMP levels following cycloheximide treatment.

The regulation of BiP by FICD-mediated AMPylation and deAMPylation provides the UPR with a rapid post-translational strand for matching the activity of a key ER chaperone to its client load. The simple biochemical mechanism proposed here for the requisite switch in FICD's antagonistic activities parallels the regulation of the UPR transducers, PERK and IRE1, whose catalytically active conformation is strictly linked to dimerisation (Dey *et al*, 2007; Lee *et al*, 2008). A simple correlation emerges, whereby ER stress favours dimerisation of UPR effectors, activating PERK and IRE1 to regulate gene expression and the FICD deAMPylase to recruit BiP into the chaperone cycle (possibly through an increased ER ADP/ATP ratio). Resolution of ER stress favours the inactive monomeric state of PERK and IRE1 and, as suggested here, the AMPylation-competent monomeric FICD (Fig 7).

# Materials and Methods

### Plasmid construction

The plasmids used in this study have been described previously or were generated by standard molecular cloning procedures and are listed in Table EV2.

### Cell lines

All cells were grown on tissue culture dishes or multi-well plates (Corning) at 37°C and 5% CO$_2$. CHO-K1 cells (ATCC CCL-61) were phenotypically validated as proline auxotrophs, and their *Cricetulus griseus* origin was confirmed by genomic sequencing. *CHOP*::*GFP* and *XBP1s*::*Turquoise* reporters were introduced sequentially under G418 and puromycin selection to generate the previously described derivative CHO-K1 S21 clone (Sekine *et al*, 2016). The cells were cultured in Nutrient mixture F-12 Ham (Sigma) supplemented with 10% (*v/v*) serum (FetalClone II; HyClone), 1 × penicillin–streptomycin (Sigma) and 2 mM L-glutamine (Sigma). The CHO-K1 *FICD$^{-/-}$* cell line used in this study was described previously (Preissler *et al*, 2015b). HEK293T cells (ATCC CRL-3216) were cultured in Dulbecco's modified Eagle's medium (Sigma) supplemented as described above. Cell lines were subjected to random testing for mycoplasma contamination using the MycoAlert Mycoplasma Detection Kit (Lonza).

Experiments were performed at cell densities of 60–90% confluence. Where indicated, cells were treated with cycloheximide (Sigma) at 100 μg/ml diluted with fresh, pre-warmed medium and then applied to the cells by medium exchange.

### Mammalian cell lysates

Cell lysis was performed as described in Preissler *et al* (2015a) with modifications. In brief, mammalian cells were cultured on 10 cm dishes and treated as indicated and/or transfected using Lipofectamine LTX with 5 μg plasmid DNA, and allowed to grow for 24–40 h. Before lysis, the dishes were placed on ice and washed with ice-cold PBS, and cells were detached in PBS containing 1 mM ethylenediaminetetraacetic acid (EDTA) using a cell scraper. The cells were sedimented for 5 min at 370 × *g* at 4°C and lysed in HG lysis buffer [20 mM HEPES-KOH pH 7.4, 150 mM NaCl, 2 mM MgCl$_2$, 10 mM D-glucose, 10% (*v/v*) glycerol, 1% (*v/v*) Triton X-100] containing protease inhibitors [2 mM phenylmethylsulphonyl fluoride (PMSF), 4 μg/ml pepstatin, 4 μg/ml leupeptin, 8 μg/ml aprotinin] with 100 U/ml hexokinase (from *Saccharomyces cerevisiae* Type F-300; Sigma) for 10 min on ice. The lysates were cleared for 10 min at 21,000 × *g* at 4°C. Bio-Rad protein assay reagent (Bio-Rad) was used to determine the protein concentrations of lysates. For analysis by SDS–PAGE, SDS sample buffer was added to the lysates and

proteins were denatured by heating for 10 min at 70°C before separation on 12.5% SDS polyacrylamide gels. To detect endogenous BiP by native-PAGE, the lysate samples were loaded immediately on native gels (see below).

## Native polyacrylamide gel electrophoresis (native-PAGE)

Non-denaturing native-PAGE was performed as described previously (Preissler *et al*, 2015a). Briefly, Tris-glycine polyacrylamide gels (4.5% stacking gel and a 7.5% separation gel) were used to separate proteins from mammalian cell lysates to detect BiP monomers and oligomers. The separation was performed in running buffer (25 mM Tris, 192 mM glycine, pH ~ 8.8) at 120 V for 2 h. Afterwards, the proteins were transferred to a polyvinylidene difluoride (PVDF) membrane in blotting buffer (48 mM Tris, 39 mM glycine; pH ~ 9.2) supplemented with 0.04 (*w/v*) SDS for 16 h at 30 V for immunodetection. The membrane was washed for 20 min in blotting buffer (without SDS) supplemented with 20% (*v/v*) methanol before blocking. Volumes of lysates corresponding to 30 μg of total protein were loaded per lane to detect endogenous BiP from cell lysates by immunoblotting.

## Streptavidin pull-down and FLAG immunoprecipitation

To analyse the formation of FICD dimers *in vivo* (Fig 1B), CHO-K1 cells were transfected with 4 μg plasmid DNA encoding His$_6$-AviTag-FICD (UK 2275) or His$_6$-AviTag-FICD$^{L258D}$ (UK 2319) and FLAG-FICD (UK 2276) or FLAG-FICD$^{L258D}$ (UK 2318), and 4 μg plasmid DNA encoding BirA [in order to keep the final amount of plasmid DNA the same, an empty pCEFL plasmid (UK 95) was used; Table EV2] as described above. 24 h before lysis, the medium was exchanged to medium containing 50 μM biotin (Molecular Probes). For streptavidin pull-down of His$_6$-AviTag-FICD, CHO-K1 cells were transfected and allowed to grow for approximately 40 h. Cells were then lysed in lysis buffer [50 mM Tris–HCl pH 7.4, 150 mM NaCl, 1% (*v/v*) Triton X-100, 10% (*v/v*) glycerol] supplemented with protease inhibitors. The lysates were cleared twice and normalised, and equal volumes of the lysates were incubated with 50 μl Dynabeads (MyOne Streptavidin C1, Life Technologies) for 60–90 min at 4°C, rotating. The beads were then recovered by centrifugation for 1 min at 200 × *g* and by placing the tube in a magnetic separation stand. The beads were washed three times at 25°C with RIPA buffer [50 mM Tris–HCl pH 8.0, 150 mM NaCl, 1% (*v/v*) Triton X-100, 0.5% (*v/v*) sodium deoxycholate, 0.1% (*v/v*) SDS] supplemented with protease inhibitors. Bound proteins were eluted in 25 μl urea sample buffer [8 M urea, 1.36% (*v/v*) SDS, 12% (*v/v*) glycerol, 40 mM Tris–HCl pH 6.8, 0.002% (*w/v*) bromophenol blue, 100 mM DTT] and heating for 10 min at 70°C. Equal volumes of the samples were loaded on a 12.5% SDS polyacrylamide gel, and His$_6$-AviTag-FICD and FLAG-FICD were detected by immunoblotting. Samples of the normalised lysates (60 μg) were loaded as an "input" control.

For the reciprocal experiment, FLAG M2 immunoprecipitation of FLAG-FICD, equal volumes of the cleared and normalised lysates were incubated with 20 μl of anti-FLAG M2 affinity gel (Sigma) for 60–90 min at 4°C, rotating. The beads were then recovered by centrifugation for 1 min at 5,000 × *g* and washed three times with RIPA buffer. The proteins were eluted in 35 μl of 2 × SDS sample

buffer (without DTT) for 10 min at 70°C. The beads were then sedimented and the supernatants were transferred to new tubes to which 50 mM DTT was added. Equal sample volumes were analysed by SDS–PAGE and immunoblotting as described above.

## Immunoblot analysis

After separation by SDS–PAGE or native-PAGE (see above), the proteins were transferred onto PVDF membranes. The membranes were blocked with 5% (*w/v*) dried skimmed milk in TBS (25 mM Tris–HCl pH 7.5, 150 mM NaCl) and incubated with primary antibodies followed by IRDye fluorescently labelled secondary antibodies (LI-COR). The membranes were scanned with an Odyssey near-infrared imager (LI-COR). Primary antibodies and antisera against hamster BiP [chicken anti-BiP (Avezov *et al*, 2013)], eIF2α [mouse anti-eIF2α (Scorsone *et al*, 1987)], FICD [chicken anti-FICD (Preissler *et al*, 2015b)], monoclonal anti-FLAG M2 (Sigma) and IRDye 800CW streptavidin (LI-COR) were used.

## Flow cytometry

FICD (wild-type and mutant) overexpression-dependent induction of unfolded protein response signalling was analysed by transient transfection of wild-type and *FICD$^{-/-}$ CHOP::GFP* CHO-K1 UPR reporter cell lines with plasmid DNA encoding the FICD protein and mCherry as a transfection marker, using Lipofectamine LTX as described previously (Preissler *et al*, 2015b). 0.5 μg DNA was used in Fig EV2B and C to transfect cells growing in 12-well plates. 40 h after transfection, the cells were washed with PBS and collected in PBS containing 4 mM EDTA, and single-cell fluorescent signals (20,000/sample) were analysed by dual-channel flow cytometry with an LSRFortessa cell analyser (BD Biosciences). GFP and mCherry fluorescence were detected with excitation laser 488 nm, filter 530/30, and excitation laser 561, filter 610/20, respectively. Data were processed using FlowJo, and median reporter (in Q1 and Q2) analysis was performed using Prism 6.0e (GraphPad).

## Production of VSV-G retrovirus in HEK293T cells and infection of CHO-K1 cells

In an attempt to establish BiP AMPylation in *FICD$^{-/-}$* cells (Fig 1A), cells were targeted with retrovirus expressing FICD (incorporating the naturally occurring repressive uORF found in its cDNA) and mCherry. HEK293T cells were split onto 6-cm dishes 24 h prior to co-transfection of pBABE-mCherry plasmid encoding FICD (UK 1939; Table EV2) with VSV-G retroviral packaging vectors, using TransIT-293 Transfection Reagent (Mirus) according to the manufacturer's instructions. 16 h after transfection, medium was changed to medium supplemented with 1% (*w/v*) BSA (Sigma). Retroviral infections were performed following a 24-h incubation by diluting 0.45-μm filter-sterilised cell culture supernatants at a 1:1 ratio into CHO-K1 cell medium supplemented with 10 μg/ml polybrene (8 ml final volume) and adding this preparation to *FICD$^{-/-}$* CHO-K1 cells (1 × 10$^6$ cells seeded onto 10 cm dishes 24 h prior to infection). Infections proceeded for 8 h, after which viral supernatant was replaced with fresh medium. 48 h later, the cells were split into four 10-cm dishes. Five days after transfection, single cells were sorted according to their mCherry intensity. Selected clones were expanded

and analysed by flow cytometry (to assess mCherry intensity) and native-PAGE (to check for BiP AMPylation).

## Protein purification

### FICD

Wild-type and mutant human FICD proteins (amino acid residues 104–445) were expressed as His$_6$-Smt3 fusion constructs in T7 Express *lysY/I$^q$* (NEB) *Escherichia coli* cells. The cells were grown in LB medium (usually 6 l per construct) containing 50 µg/ml kanamycin at 37°C to an optical density (OD$_{600\ nm}$) of 0.6 and then shifted to 18°C for 20 min, followed by induction of protein expression with 0.5 mM isopropylthio β-D-1-galactopyranoside (IPTG). The cultures were further incubated for 16 h at 18°C, harvested and lysed with a high-pressure homogeniser (EmulsiFlex-C3; Avestin) in buffer A [25 mM Tris–HCl pH 8.0, 500 mM NaCl, 40 mM imidazole, 1 mM MgCl$_2$, 0.1 mM tris(2-carboxyethyl)phosphine (TCEP)] containing protease inhibitors (2 mM PMSF, 4 µg/ml pepstatin, 4 µg/ml leupeptin, 8 µg/ml aprotinin), 0.1 mg/ml DNase I and 20 µg/ml RNaseA. The lysates were centrifuged for 30 min at 45,000 × g and incubated with 1 ml of Ni-NTA agarose (Qiagen) per 1 l expression culture, for 30 min rotating at 4°C. Afterwards, the beads were transferred to a gravity-flow Econo column (49 ml volume; Bio-Rad) and washed with five column volumes (CV) buffer A without MgCl$_2$ and buffer B (25 mM Tris–HCl pH 8.0, 150 mM NaCl, 10 mM imidazole, 0.1 mM TCEP), respectively. The beads were further washed with buffer B sequentially supplemented with (i) 1 M NaCl, (ii) 10 mM MgCl$_2$ + 5 mM ATP and (iii) 0.5 M Tris–HCl pH 8.0 (each 5 CV), followed by 2 CV TNT-Iz10 (25 mM Tris–HCl pH 8.0, 150 mM NaCl, 1 mM TCEP, 10 mM imidazole). Proteins were eluted by on-column cleavage with 1.5 µg/ml Ulp1 protease carrying a C-terminal StrepII-tag [Ulp1-StrepII (UK 1983)] in 1 bed volume TNT-Iz10 overnight at 4°C. The eluate was collected, retained cleavage products were washed off the beads with TNT-Iz10, and all fractions were pooled. The total eluate was diluted 1:2 with 25 mM Tris–HCl pH 8.0 and further purified by anion-exchange chromatography using a 6 ml RESOURCE Q column (GE Healthcare) equilibrated in 95% AEX-A (25 mM Tris–HCl pH 8.0, 25 mM NaCl) and 5% AEX-B (25 mM Tris–HCl, 1 M NaCl). Proteins were eluted by applying a gradient from 5 to 30% AEX-B in 20 CV at 3 ml/min. Fractions of elution peaks (absorbance at 280 nm, A$_{280\ nm}$) corresponding to monomeric or dimeric FICD were pooled and concentrated using 30-kDa MWCO centrifugal filters (Amicon Ultra; Merck Millipore) in the presence of 1 mM TCEP. The proteins were then subjected to size-exclusion chromatography using a HiLoad 16/60 Superdex 200 prep grade column (GE Healthcare) equilibrated in SEC buffer (25 mM Tris–HCl pH 8.0, 150 mM NaCl). Peaks corresponding to monomeric or dimeric FICD were supplemented with 1 mM TCEP, concentrated (> 120 µM) and frozen in aliquots.

### BiP

Mutant Chinese hamster BiP proteins with an N-terminal His$_6$-tag were purified as described before with modifications (Preissler *et al*, 2017b). Note, hamster and human BiP are essentially identical in terms of amino acid identity outside of the (cleaved) signal sequence with a single amino acid difference (Ala650Ser from human to hamster) present in the unstructured C terminus two amino acids

from the terminal KDEL. Proteins were expressed in M15 *E. coli* cells (Qiagen). The bacterial cultures were grown in LB medium supplemented with 100 µg/ml ampicillin and 50 µg/ml kanamycin at 37°C to an OD$_{600\ nm}$ of 0.8 and expression was induced with 1 mM IPTG. The cells were further grown for 6 h at 37°C, harvested and lysed in buffer C [50 mM Tris–HCl pH 8, 500 mM NaCl, 1 mM MgCl$_2$, 10% (*v/v*) glycerol, 20 mM imidazole] containing 0.1 mg/ml DNaseI and protease inhibitors. The lysates were cleared for 30 min at 45,000 × g and incubated with 1 ml of Ni-NTA agarose (Qiagen) per 1 l of expression culture for 2 h rotating at 4°C. Afterwards, the matrix was transferred to a gravity-flow Econo column (49 ml volume; Bio-Rad) and washed with buffer D (50 mM Tris–HCl pH 8.0, 500 mM NaCl, 10% (*v/v*) glycerol, 30 mM imidazole), buffer E (50 mM Tris–HCl pH 8.0, 300 mM NaCl, 10 mM imidazole, 5 mM β-mercaptoethanol) and buffer E sequentially supplemented with (i) 1 M NaCl, (ii) 10 mM MgCl$_2$ + 3 mM ATP, (iii) 0.5 M Tris–HCl pH 7.4 or (iv) 35 mM imidazole. The BiP proteins were then eluted with buffer F (50 mM Tris–HCl pH 8.0, 300 mM NaCl, 5 mM β-mercaptoethanol, 250 mM imidazole), dialysed against HKM (50 mM HEPES-KOH pH 7.4, 150 mM KCl, 10 mM MgCl$_2$) and concentrated with 30-kDa MWCO centrifugal filters. The proteins were flash-frozen in aliquots and stored at −80°C.

GST-TEV-BiP constructs were purified like His$_6$-Smt3-FICD, above, with minor alterations. Purification was performed without imidazole in the purification buffers. Cleared lysates were supplemented with 1 mM DTT and incubated with GSH-Sepharose 4B matrix (GE Healthcare) for 1 h at 4°C. 2 CV of TNT(0.1) (25 mM Tris–HCl pH 8.0, 150 mM NaCl, 0.1 mM TCEP) was used as a final wash step before elution. GST-TEV-BiP was eluted with 10 mM HEPES-KOH pH 7.4, 20 mM Tris–HCl pH 8.0, 30 mM KCl, 120 mM NaCl, 2 mM MgCl$_2$ and 40 mM reduced glutathione. The eluted proteins were cleaved with 1/200 (w/w) TEV protease (UK 759), whilst dialysing into TN (25 mM Tris–HCl pH 8.0, 150 mM NaCl) plus 1 mM DTT for 16 h at 4°C. Uncleaved BiP was depleted by incubation with GSH-Sepharose 4B matrix (1 ml per 5 mg of protein) for 1 h at 4°C. The flow through was collected. Retained, cleaved material was washed from the matrix with 5 CV of TNT (0.1) and pooled. In order to AMPylate BiP, the cleaved product was combined with 1/50 (*w/w*) GST-TEV-FICD$^{E234G}$ (UK 1479; purified like the GST-TEV-BiP without the TEV cleavage steps). The AMPylation reaction was supplemented with 10 mM MgATP (10 mM MgCl$_2$ + 10 mM ATP) and incubated for 16 h at 25°C. GST-TEV-FICD was then depleted by incubation with GSH-Sepharose 4B matrix as described above. Proteins were concentrated to > 200 µM. Aliquots were snap-frozen in liquid nitrogen and stored at −80°C.

### Formation of disulphide-linked FICD dimers

Expression and purification of disulphide-linked dimers [of FICD$^{A252C-C421S}$ (UK 2219) and FICD$^{A252C-H363A-C421S}$ (trap; UK 2269)] were performed as described above with some alterations. After the affinity chromatography step, on-column cleavage was performed in TNT-Iz10 containing 1.5 µg/ml Upl1-StrepII and the retained cleavage products were washed off the beads with TN-Iz10 (25 mM Tris–HCl pH 8.0, 150 mM NaCl, 10 mM imidazole) in the absence of reducing agent. The pooled eluate was concentrated and diluted 1:4 with TN-Iz10. To allow for efficient disulphide bond formation, the samples were supplemented with 20 mM oxidised glutathione and incubated overnight at 4°C. Afterwards, the protein

solutions were diluted 1:2 with 25 mM Tris–HCl pH 8.0 and further purified by anion-exchange and size-exclusion chromatography. The final preparations were analysed by non-reducing SDS–PAGE to confirm quantitative formation of covalently linked dimers (> 95%). Cysteine-free FICD$^{C421S}$ (UK 2161) was purified according to the same protocol. A separate preparation of non-disulphide-bonded FICD$^{A252C-C421S}$ (UK 2219), which was not subjected to oxidation with glutathione, was used in control experiments (Fig 3B, Appendix Fig S1A and C).

### *In vitro* AMPylation

Standard radioactive *in vitro* AMPylation reactions were performed in HKMC buffer (50 mM HEPES-KOH pH 7.4, 150 mM KCl, 10 mM MgCl$_2$, 1 mM CaCl$_2$) containing 40 µM ATP, 0.034 MBq [α-$^{32}$P]-ATP (EasyTide; Perkin Elmer), 0.2 µM FICD, and 1.5 µM ATP hydrolysis and substrate-binding deficient BiP$^{T229A-V461F}$ (UK 1825) in a final volume of 15 µl. Where indicated, samples contained 5 µM $_{S-S}$FICD$^{A252C-H363A-C421S}$ (UK 2269, trap) to sequester modified BiP. The reactions were started by addition of nucleotides. After 20-min incubation at 25°C, the reactions were stopped by addition of 5 µl 4 × SDS sample buffer and denaturation for 5 min at 75°C. The samples were applied to SDS–PAGE and the gels were stained with Coomassie (InstantBlue; Expedeon). The dried gels were exposed to a storage phosphor screen, and radioactive signals were detected with a Typhoon biomolecular imager (GE Healthcare). Signals were quantified using ImageJ64 software (NIH).

The reactions to analyse AMPylation at elevated FICD concentrations (2 or 10 µM; Fig EV3A) contained 2 µM BiP$^{T229A-V461F}$, 80 µM ATP and 0.034 MBq [α-$^{32}$P]-ATP in a final volume of 15 µl. The reactions were stopped after 5-min incubation at 25°C.

Time course experiments (Fig 2E) were performed likewise but reactions contained 40 µM ATP, 0.136 MBq [α-$^{32}$P]-ATP, 0.3 µM FICD, 2 µM BiP$^{T229A-V461F}$ and 5 µM trap in a final volume of 60 µl. The reactions were incubated at 30°C, and samples (15 µl) were taken at different time intervals and processed as described above.

To study the effect of the concentration of wild-type FICD protein on its ability to establish a pool of AMPylated BiP (Fig 3A), final reactions were set up with 400 µM ATP, 0.049 MBq [α-$^{32}$P]-ATP, 2.5–400 nM FICD (UK 2052) and 5 µM BiP$^{T229A-V461F}$, without or with 5 µM trap in a final volume of 15 µl. The reactions were pre-incubated for 2 h before addition of nucleotides. After 2- and 16-h incubation with nucleotides at 25°C (as indicated), samples (5 µl) were taken and denatured by heating in SDS sample buffer for analysis.

To compare the activity of disulphide-bonded FICD under non-reducing and reducing conditions (Fig 3C), $_{S-S}$FICD$^{A252C-C421S}$ protein (UK 2219) was pre-incubated 16 h at 25°C without or with 10 mM DTT and a sample was analysed by non-reducing SDS–PAGE after denaturation in SDS sample buffer containing 40 mM N-ethylmaleimide (NEM). Afterwards, AMPylation reactions (15 µl final volume) were set up with 400 µM ATP, 0.049 MBq [α-$^{32}$P]-ATP, 2.5–400 nM $_{S-S}$FICD$^{A252C-C421S}$ and 5 µM BiP$^{T229A-V461F}$ in the presence or absence of 5 mM DTT. Samples were incubated for 16 h at 25°C and 5 µl was taken and processed for analysis by reducing SDS–PAGE as described above. Parallel reactions performed with cysteine-free FICD$^{C421S}$ (UK 2161), which underwent the same purification and oxidation procedure, served as a control. The experiment presented in Appendix Fig S1C was performed accordingly under non-reducing

conditions, but the reactions were incubated for 2 h at 25°C and in the presence of 5 µM trap.

### Coupled *in vitro* AMPylation/deAMPylation reactions

To measure AMPylation-/deAMPylation-dependent AMP production by FICD proteins, reactions were set up in HKM buffer containing 250 µM ATP, 0.0185 MBq [α-$^{32}$P]-ATP, 3 mM TCEP, 5 µM ATP hydrolysis-deficient BiP$^{T229A}$ (UK 838) and 2 µM FICD variant in a final volume of 30 µl. The reactions were started by addition of nucleotides and incubated for 2 h at 30°C. Afterwards, 2 µl was spotted onto a thin-layer chromatography (TLC) plate (PEI Cellulose F; Merck Millipore) pre-spotted with 2 µl of nucleotide mix containing AMP, ADP and ATP (each at 3.5 mM). The TLC plate was developed with 400 mM LiCl and 10% (*v/v*) acetic acid as a mobile phase and the dried plates were exposed to a storage phosphor screen. The signals were detected with a Typhoon biomolecular imager and quantified using ImageJ64.

### DeAMPylation measured by fluorescence polarisation (FP)

Measurement of deAMPylation kinetics was performed as described previously (Preissler *et al*, 2017a) with modifications. The probe (BiP$^{V461F}$ modified with fluorescent, FAM-labelled AMP; BiP$^{V461F}$-AMP$^{FAM}$) was generated by pre-incubating FICD$^{E234G}$ at 25 µM in HKM buffer with 200 µM ATP-FAM [N$^6$-(6-amino)hexyl-adenosine-5′-triphosphate; Jena Bioscience] for 10 min at 30°C, followed by addition of 25 µM His$_6$-tagged BiP$^{V461F}$ (UK 182) to a final volume of 50 µl, and further incubation for 2 h at 30°C. Afterwards, the reaction was diluted with 950 µl of HKMG-Iz20 [50 mM HEPES-KOH pH 7.4, 150 mM KCl, 10 mM MgCl$_2$, 5% (*v/v*) glycerol, 20 mM imidazole] and BiP proteins were bound to 80 µl Ni-NTA agarose beads (Qiagen) for 30 min at 25°C in the presence of 0.01% Triton X-100. Following several wash steps in the same buffer, proteins were eluted in HKMG-Iz250 [50 mM HEPES-KOH pH 7.4, 150 mM KCl, 10 mM MgCl$_2$, 5% (*v/v*) glycerol, 250 mM imidazole], flash-frozen in aliquots and stored at −80°C.

DeAMPylation reactions were performed in FP buffer [50 mM HEPES-KOH pH 7.4, 150 mM KCl, 10 mM MgCl$_2$, 1 mM CaCl$_2$, 0.1% (*v/v*) Triton X-100] in 384-well polystyrene microplates (black, flat bottom, µCLEAR; greiner bio-one) at 30°C in a final volume of 30 µl containing trace amounts of fluorescent BiP$^{V461F-AMP-FAM}$ probe (17 nM) and FICD proteins (0.75 or 7.5 µM). Fluorescence polarisation of FAM ($\lambda_{ex}$ = 485 nm, $\lambda_{em}$ = 535 nm) was measured with an Infinite F500 plate reader (Tecan). Fitting of the raw data to a single-exponential decay function was done using Prism 6.0e (GraphPad).

### Analytical size-exclusion chromatography

Analytical size-exclusion chromatography (SEC) was performed as described previously (Preissler *et al*, 2015a). Purified FICD proteins were adjusted to 20 µM in HKMC buffer (50 mM HEPES-KOH pH 7.4, 150 mM KCl, 10 mM MgCl$_2$, 1 mM CaCl$_2$) and incubated at 25°C for at least 20 min before injection. 10 µl of each sample was injected onto a SEC-3 HPLC column (300 Å pore size; Agilent Technologies) equilibrated with HKMC at a flow rate of 0.3 ml/min. Runs were performed at 25°C, and A$_{280\ nm}$ absorbance traces were recorded. Protein standards (Bio-Rad, cat. no. 151–1,901) were run

as size references, and the elution peaks of γ-globulin (158 kDa), ovalbumin (44 kDa) and myoglobin (17 kDa) are indicated. For dimer SEC studies in Fig EV1D and E, the FICD proteins were incubated for 16 h at 25°C before injection. In Fig EV2H, in order to investigate by SEC capture of AMPylated BiP by $_{S-S}$FICD$^{A252C-H363A-C421S}$ (UK 2269, trap) *in vitro* AMPylation reactions containing different combinations of 20 μM BiP$^{T229A-V461F}$ (UK 1825), 10 μM trap and 3 μM FICD$^{L258D}$ (UK 2091) were performed in HKMC (supplemented with 2 mM ATP where indicated) and incubated for 1.5 h at 30°C before injection.

### Fluorescence detection system sedimentation velocity analytical ultracentrifugation (FDS-SV-AUC)

Bacterial expression and purification of FICD proteins carrying an N-terminal cysteine for site-specific labelling (FICD$^{NC}$, UK 2339, and FICD$^{L258D-NC}$, UK 2367) were performed as described above with the following alterations: Cells were lysed in the presence of 5 mM β-mercaptoethanol and the eluate pool after affinity chromatography and on-column cleavage was supplemented with 5 mM DTT and diluted 1:2 with 25 mM Tris–HCl pH 8.0 containing 0.2 mM TCEP. The subsequent anion-exchange chromatography step was performed with buffer solutions AEX-A and AEX-B supplemented with 0.2 mM TCEP. Afterwards, the peak fractions corresponding to the dimeric form of FICD were pooled and concentrated. The protein at 200 μM was labelled in a 150 μl reaction volume with 600 μM Oregon Green 488-iodoacetamide in the presence of 0.5 mM TCEP and 0.1 mM EDTA for 16 h at 4°C. The reaction was quenched with 2 mM DTT for 10 min at 25°C. Afterwards, the sample was passed through a CentriPure P2 desalting column (emp) equilibrated in SEC buffer containing 0.2 mM TCEP. The eluate was applied to size-exclusion chromatography using a Superdex 200 10/300 GL column (GE Healthcare) in the presence of 0.2 mM TCEP. The fractions of the $A_{280\ nm}$ peak, corresponding to dimeric FICD, were pooled, and the concentration of TCEP was adjusted to 1 mM. The proteins were concentrated and frozen in aliquots. The protein concentration was determined after denaturing the proteins with 6 M guanidine hydrochloride by measuring absorbance at 280 and 496 nm with a NanoDrop Spectrophotometer (Thermo Fisher Scientific). The concentration was calculated using the following equation:

$$\text{Protein concentration (M)} = [A_{280\ nm} - (A_{496\ nm} \times 0.12)]/\varepsilon$$

where 0.12 is the correction factor for the fluorophore's absorbance at 280 nm, and ε is the calculated molar extinction coefficient of FICD (29,340 cm$^{-1}$ M$^{-1}$). The labelling efficiency of the FICD$^{NC}$ preparation was 74% as calculated based on the $A_{496\ nm}$ value and assuming an extinction coefficient for Oregon Green 488 of 70,000 cm$^{-1}$ M$^{-1}$. The labelling efficiency of the monomeric FICD$^{L258D-NC}$ control preparation was 9.6%. Labelling of the endogenous cysteine residue (Cys421) of wild-type FICD was very inefficient (< 1%) and thus considered negligible.

Samples of Oregon Green-labelled FICD in 50 mM HEPES-KOH pH 7.4, 150 mM KCl, 10 mM MgCl$_2$, 1 mM CaCl$_2$, 0.3 mM TCEP, 0.1% Tween-20, 0.15 mg/ml BSA (Sigma), ranging in concentration from 1.6 μM to 31 pM, were centrifuged at 45,000 rpm at 20°C in an An50Ti rotor using an Optima XL-I analytical ultracentrifuge

(Beckmann) equipped with a fluorescence optical detection system (Aviv Biomedical) with fixed excitation at 488 nm and fluorescence detection at > 505 nm. Data were processed and analysed using SEDFIT 15 and SEDPHAT 13b (Schuck, 2003) according to the published protocol for high-affinity interactions detected by fluorescence (Chaturvedi *et al*, 2017). Data were plotted with Prism 6.0e (GraphPad) or GUSSI (Brautigam, 2015).

### Differential scanning fluorimetry

Differential scanning fluorimetry experiments were performed on an ABi 7500 qPCR machine (Applied Biosciences). Experiments were carried out in 96-well qPCR plates (Thermo Fisher), with each sample in technical triplicate and in a final volume of 20 μl. Protein was used at a final concentration of 2 μM, ligands at the indicated concentrations (2.5–20 mM) and SYPRO Orange dye (Thermo Fisher) at a 10 × concentration in a buffer of HKM plus 1 mM TCEP (unless otherwise specified). For the ATP titration (Figs 4F and EV3E), the DSF buffer was supplemented with an additional 15 mM MgCl$_2$ (25 mM total MgCl$_2$). For Figs 4F and EV3E, no protein control (NPC) wells including either no nucleotide or 20 mM ATP appeared indistinguishable and these values were averaged for calculation of the fluorescence background. For Figs 4E and EV3D, background fluorescence was calculated separately for each nucleotide condition using NPC wells with either no nucleotide, 5 mM ATP or 2.5 mM ADP. Fluorescence of the SYPRO Orange dye was monitored over a temperature range of 20–95°C using the VIC filter set. Data were then analysed in Prism 7.0e (GraphPad), with melting temperature calculated as the global minimums of the negative first derivatives of the relative fluorescent unit (RFU) melt curves (with respect to temperature).

### Protein crystallisation and structure determination

FICD proteins were purified as above in *Protein Purification* but gel-filtered into a final buffer [T(10)NT] of 10 mM Tris–HCl pH 8.0, 150 mM NaCl, and 1 mM TCEP. Proteins were diluted to 9 mg/ml in T(10)NT prior to crystallisation, via sitting drop vapour diffusion. For structures containing ATP/AMPPNP, protein solutions were supplemented with MgATP/AMPPNP (from a pH 7.4, 100 mM stock solution) to a final concentration of 10 mM. A drop ratio of protein solution to crystallisation well solution of 200:100 nl was used. Where applicable, crystals were obtained by microseeding (D'Arcy *et al*, 2007), from conditions provided in Table EV1. In these instances, a drop ratio of protein solution to water-diluted seeds to crystallisation well solution of 150:50:100 nl was used. The best diffracting crystals were obtained from the crystallisation conditions detailed in Table EV1, cryoprotected by briefly soaking in cryoprotectant solution (Table EV1) and then cryocooled in liquid nitrogen.

Diffraction data were collected from the Diamond Light Source, and the data processed using XDS (Kabsch, 2010) and the CCP4 module Aimless (Winn *et al*, 2011; Evans & Murshudov, 2013). Structures were solved by molecular replacement using the CCP4 module Phaser (McCoy *et al*, 2007; Winn *et al*, 2011). For the FICD$^{L258D}$:Apo and FICD:ATP structures, the human FICD protein structure 4U0U (FICD:MgADP) from the Protein Data Bank (PDB)

was used as a search model. Subsequent molecular replacements used the solved FICD$^{L258D}$:Apo structure as a search model. Manual model building was carried out in COOT (Emsley *et al*, 2010) and refined using refmac5 (Winn *et al*, 2003). Metal binding sites were validated using the CheckMyMetal server (Zheng *et al*, 2017). Polder (OMIT) maps were generated using the Polder Map module of Phenix (Adams *et al*, 2010; Liebschner *et al*, 2017). Structural figures were prepared using UCSF Chimera (Pettersen *et al*, 2004) and PyMol (Schrödinger, LLC, 2015).

## BioLayer interferometry

### In vitro *biotinylation*

Ligands for BLI were generated from the tag-cleaved forms of unmodified or AMPylated GST-TEV-AviTag-haBiP$^{V461F}$ (UK 2043) and GST-TEV-AviTag-haBiP$^{T229A-V461F}$ (UK 2331). Biotinylation was conducted *in vitro* with 100 μM target protein, 200 μM biotin (Sigma) and 2 μM GST-BirA (UK 1801) in a buffer of 2 mM ATP, 5 mM MgCl$_2$, 25 mM Tris–HCl pH 8.0, 150 mM NaCl and 1 mM TCEP. The reaction mixture was incubated for 16 h at 4°C. Excess biotin was removed by size-exclusion chromatography on a S200 10/300 GL column (GE Healthcare) with a distal 1 ml GSTrap 4B (GE Healthcare), connected in series. The ligand was confirmed as being > 95% biotinylated as judged by streptavidin gel-shift. In the case of biotinylated-AviTag-haBiP$^{T229A-V461F}$, this protein was also made nucleotide-free by the addition of 2 U calf intestinal alkaline phosphatase (NEB) per mg of BiP, plus extensive dialysis into TN buffer with 1 mM DTT and 2 mM EDTA (dialysed with several dialysate changes, for 2 days at 4°C). The protein was then purified by anion-exchange chromatography on a MonoQ 5/50 GL column (GE Healthcare) using buffers AEX-A and AEX-B with a gradient of 7.5–50% B over 20 CV at a flow rate of 1 ml/min. The protein was concentrated using a 30 kDa MWCO centrifugal filters (Amicon Ultra; Merck Millipore) and then gel-filtered, as above, but into an HKM buffer. Fractions were pooled and supplemented with 1 mM TCEP. All proteins after biotinylation and purification were concentrated to > 20 μM, flash-frozen in small aliquots and stored at −80°C.

### Kinetic experiments

All BLI experiments were conducted on the FortéBio Octet RED96 System (Pall FortéBio) using an HKM plus 0.05% Triton X-100 (HKMTx)-based buffer system. Nucleotide was added as indicated. Streptavidin (SA)-coated biosensors (Pall FortéBio) were hydrated in HKMTx for at least 30 min prior to use. Experiments were conducted at 30°C. BLI reactions were prepared in 200 μl volumes in 96-well microplates (greiner bio-one). Ligand loading was performed for 300–600 s at a shake speed of 1,000 rpm until a binding signal of 1 nm was reached. The immobilised ligand sensor was then baselined in assay solution for at least 200 s. For kinetic experiments with biotinylated-AviTag-haBiP$^{T229A-V461F}$:Apo [BiP$^{T229A-V461F}$:Apo (UK 2331)] loaded on the tip, a 10 Hz acquisition rate was used and the baseline, association and dissociation steps were conducted at a 400 rpm shake speed. Preceding the baseline step, biotinylated BiP$^{T229A-V461F}$:Apo was also activated with or without 2 mM ATP (unless otherwise stated), as indicated, for 300 s at a 1,000 rpm shake speed. In these experiments, FICD analyte association or dissociation steps were conducted in the

presence or absence of nucleotide, as indicated, with ATP at 8 mM and ADP at 2 mM. These concentrations were chosen in an attempt to saturate either monomeric or dimeric FICD with the respective nucleotide [$K_d$ of MgADP for wild-type FICD is 1.52 μM by ITC (Bunney *et al*, 2014); $K_{1/2}$ of ATP induced FICD $T_m$ shift in the low mM range] and/or to make ATP binding non-rate limiting. In Fig EV6A, as a control for the absence of substantial ATP dissociation from BiP, between the activation and baseline step, an additional 1,500 s BiP wash (± ATP as indicated) was included. Other BLI experiments were conducted with all steps at a 1,000 rpm shake speed with a 5 Hz acquisition rate. All association–dissociation kinetics were completed in ≤ 1,500 s. Data were processed in Prism 7.0e (GraphPad). Note, the FICD variants used as analytes in all BLI experiments were catalytically inactive His363Ala variants (used at 250 nM).

In the dimer dissociation BLI experiments (Fig 6E, and Fig EV6E and F) biotinylated-AviTag-FICD$^{H363A}$ (UK 2422) was diluted to 3 nM and incubated for 10 min at 25°C with either dimeric FICD$^{H363A}$ or monomeric FICD$^{L258D-H363A}$ (at 300 nM) in HKMTx. After this incubation period, the streptavidin biosensors were loaded until the hetero-labelled dimers (biotinylated-AviTag-FICD$^{H363A}$ with FICD$^{H363A}$) reached 1 nm displacement. Dissociation was initiated by dipping in HKTx buffer (50 mM HEPES-KOH pH 7.4, 150 mM KCl and 0.05% Triton X-100) ± nucleotide at 5 mM, as indicated. Data were processed by subtracting the respective monomer incubated biotinylated FICD sensor signals from the dimeric hetero-labelled dimer dissociations, followed by fitting of the corrected dissociations to mono-exponential decays using Prism 7.0e (GraphPad).

## Fluorescence-based FICD monomer-dimer assay

To generate an oligomeric state-sensitive fluorescent FICD probe, we used the dimerisation-competent, cysteine-free, catalytically inactive FICD$^{H363A-C421S}$ and introduced cysteines at positions of surface-exposed residues predicted to lie within a FRET distance across the dimer interface and decorated them with fluorophores, exploiting the reactivity of the introduced cysteine. Residues on the N-terminal extension of the Fic domain (Thr277Cys, Ser241Cys, Lys256Cys) were either very poorly labelled or upon labelling destabilised the protein. Stable FICD oligomeric state-sensitive fluorescent probes were engineered by the introduction of cysteine substitutions into the surface of the core Fic domain (Ser288Cys and Arg308Cys, with the former proving most useful for the following experimental setting). The Ser288Cys labelling location was selected on the basis of it being a solvent exposed residue, within an α-helical region of the core Fic domain and close to the dimer interface. Moreover, the original serine side chain was not observed to be engaged in inter-residue contacts. The resulting construct (UK 2473; FICD$^{S288C-H363A-C421S}$) was expressed and Ulp1-cleaved, as detailed above, in reducing conditions. The protein was then subject to site-specific labelling of the cysteine, using tetramethylrhodamine-maleimide [TMR-maleimide, a fluorophore chosen based on its narrow Stokes shift and therefore well suited for homo-FRET (Pietraszewska-Bogiel & Gadella, 2011; Yang *et al*, 2017)]. In brief, 100 μM protein was mixed with 1 mM TMR-5-maleimide (Sigma) in a final solution composed of 50 mM HEPES pH 7.2, 150 mM KCl, 0.5 mM TCEP, 1 mM EDTA and 4% DMF. The 250 μl reaction

mixture was incubated for 16 h at 4°C in the dark. The resulting labelled protein was first buffer exchanged into HKT(0.2) (50 mM HEPES-KOH pH 7.4, 150 mM KCl, 0.2 mM TCEP) using a Centri Pure P2 desalting column (emp Biotech). Any unlabelled cysteines in the eluted protein were then quenched by incubation with 1 mM NEM for 1 h at 4°C, followed by incubation with 5 mM DTT for a further 1 h at 4°C to quench excess NEM. The resulting protein was diluted 1:2 with 25 mM Tris pH 8.0 and loaded onto a Mono Q 5/50 GL column (GE Healthcare). The labelled protein was eluted with a linear gradient of AEX-A (plus 0.2 mM TCEP) to AEX-B (plus 0.2 mM TCEP) of 5–35% over 30 CV. Labelled protein fractions were pooled and concentrated using a 30 kDa MWCO centrifugal filter (Amicon Ultra; Merck Millipore). The concentrated labelled protein was then buffer exchanged into 25 mM HEPES-KOH pH 7.4, 50 mM KCl, 100 mM NaCl and 2 mM $MgCl_2$ and confirmed as being dimerisation competent, using an S200 Increase 10/300 GL column (GE Healthcare). Protein fractions were pooled and concentrated. The resulting protein concentration and TMR labelling efficiency were estimated by measurement of the $A_{280\,nm}$ and $A_{520\,nm}$ using a NanoDrop Spectrophotometer (Thermo Fisher Scientific). The protein concentration was calculated using the following equation:

$$\text{Protein concentration (M)} = [A_{280\,nm} - (A_{555\,nm} \times 0.30)]/\varepsilon$$

where 0.30 is the correction factor for the fluorophore's absorbance at 280 nm, and $\varepsilon$ is the calculated molar extinction coefficient of FICD ($29{,}340$ $cm^{-1}$ $M^{-1}$). The labelling efficiency of the preparation was 41% as calculated based on the $A_{555\,nm}$ value and assuming an extinction coefficient for TMR of $65{,}000$ $cm^{-1}$ $M^{-1}$.

For the fluorescence assay, FICD[S288C-H363A-C421S]-TMR was diluted to 2.5 nM and mixed with the titrant, as indicated, in a buffer of HKTx dispensed into 384-well non-binding, low volume, HiBase, black microplates (greiner bio-one). Note, all FICD variants were dispensed directly into the microplate well solutions using a D300e digital dispenser (Tecan). The plate was then sealed and the final 20 μl (Fig 6F) or 15 μl reactions (Fig 6G) were incubated for 45 min at 10°C, whilst shaking at 300 rpm on a ThermoMixer C (Eppendorf). The plate seal was then removed and fluorescent measurements were conducted using a CLARIOstar Plus plate reader (BMG Labtech), exciting at 535–20 nm and top reading emission at 585–30 nm. For each condition, a reference well lacking FICD[S288C-H363A-C421S]-TMR was included. This background fluorescence was subtracted from the respective condition with FICD[S288C-H363A-C421S]-TMR. In order to correct for any non-specific effects of the nucleotide titrants on TMR fluorescence, each fluorescence value in Fig 6G is presented as a fraction of the de-quenched "monomeric" FICD-TMR signal from a parallel sample containing an additional 250 nM FICD[H363A] (UK 1954). Independent repeat data were collected in technical duplicate.

The displayed best-fit lines were derived by non-linear regression fitting of a one-site binding saturation model using Prism 7.0e (GraphPad).

## Data availability

The FICD crystal structures have been deposited in the Protein Data Bank (PDB) with the following accession codes: 6I7G; https://www.rcsb.org/structure/6I7G (FICD:ATP), 6I7H; https://www.rcsb.org/structure/6I7H (FICD[K256S]:Apo), 6I7I; https://www.rcsb.org/structure/6I7I (FICD[K256A]:MgATP), 6I7J; https://www.rcsb.org/structure/6I7J (FICD[L258D]:Apo), 6I7K; https://www.rcsb.org/structure/6I7K (FICD[L258D]:MgATP) and 6I7L; https://www.rcsb.org/structure/6I7L (FICD[L258D]:MgAMPPNP).

**Expanded View** for this article is available online.

## Acknowledgements

We thank the Huntington lab for access to the Octet machine, Claudia Flandoli for scientific illustrations and the CIMR flow cytometry core facility team (Reiner Schulte, Chiara Cossetti and Gabriela Grondys-Kotarba). We also thank the Diamond Light Source for beamtime (proposal mx15916) and the staff of beamlines I03 and I04 for assistance with data collection. This work was supported by Wellcome Trust Principal Research Fellowship to D.R. (Wellcome 200848/Z/16/Z), Medical Research Council PhD programme funding to L.A.P. (MR/K50127X/1), a Wellcome Trust Principal Research Fellowship to R.J.R. (Wellcome 082961/Z/07/Z) and a Wellcome Trust Strategic Award to the Cambridge Institute for Medical Research (Wellcome 100140).

## Author contributions

LAP co-led and conceived the project, designed and conducted the biophysical experiments, analysed and interpreted the data, purified and crystallised proteins, collected, analysed and interpreted the X-ray diffraction data, and wrote the article. CR designed, conducted and interpreted the *in vivo* experiments and contributed to revising the article. YY supervised crystallisation efforts as well as the collection and processing of the X-ray diffraction data, contributed to analysis and interpretation of the structural data and to revising the article. LN contributed to the *in vivo* experiments. SHM conducted the AUC experiments and analysed the AUC data, and contributed to revising the article. RJR contributed to analysis and interpretation of the structural data and to revising the article. SP co-led and conceived the project, designed and conducted the biochemical experiments, analysed and interpreted the data, and wrote the article. DR conceived and oversaw the project, interpreted the data, and wrote the article.

## Conflict of interest

The authors declare that they have no conflict of interest.

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
