## [Review Process File · The EMBO Journal]

An oligomeric state-dependent switch in the ER enzyme FICD regulates AMPylation and deAMPylation of BiP

Luke A. Perera, Claudia Rato, Yahui Yan, Lisa Neidhardt, Stephen H. McLaughlin, Randy J. Read, Steffen Preissler, David Ron.

Review timeline:	Submission date:	4 th April 2019
	Editorial Decision:	8 th May 2019
	Revision received:	10 th June 2019
	Editorial Decision:	3 rd July 2019
	Revision received:	14 th July 2019
	Editorial Decision:	19 th July 2019
	Revision received:	25 th July 2019
	Accepted:	26 th July 2019

Editor: Stefanie Boehm

Transaction Report:

1st Editorial Decision

8th May 2019

Thank you for submitting your manuscript on FCID regulation of BiP AMPylation for consideration by The EMBO Journal. We have now received three referee reports on your study, which are included below for your information.

As you will see, the reviewers are overall positive, but also raise some concerns that would need to be addressed in a revised manuscript. Referee #2 in particular is not convinced that the role of ATP has been decisively proven and raises several experimental points.

Should you be able to adequately address the key concerns then we would be happy to consider this study further for publication. I would therefore like to invite you to prepare and submit a revised manuscript.

REFeree REPORTS

Referee #1:

The manuscript by Perera et al. provides several lines of evidence defining the basis for the switch in FICD's activity between AMPylation and deAMPylation of the ER chaperone BiP, critical for UPR. Specifically, the transition of FICD from a dimeric enzyme, that deAMPylates BiP, to a monomer with potent AMPylation activity is mainly demonstrated using relevant FICD variants in vitro and in cells. Experimental evidence fully supports the main conclusions.

One important aspect of the proposed model in Figure 7 is the identity of cellular regulators that affect FICD monomer/dimer transitions and in turn AMPylation/ deAMPylation. It is speculated that enzymatic activity might be regulated by the ADP/ATP ratio in the ER, based on limited data in vitro. It could be better to include in this discussion other possibilities (other changes in the ER environment, including other small molecules and interaction partners). It is understandable that further supporting data will be another major undertaking.

Referee #2:

The manuscript addresses the mechanisms that determine whether BiP is ampylated or de-ampylated by FICD. Ampylation is inhibitory of BiP and occurs under specific conditions, such as recovery from stress. FICD is a homodimer that de-ampylates BiP, and the ampylation reaction by FICD is observed when an inhibitory Glu234 is mutated. The manuscript concludes that monomerization of FICD allows ampylation activity, by changing an allosteric network of bonds that position Glu234. High resolution structures suggest that dimerization affects ATP binding, and ATP in turn is found to affect monomerization and BiP binding. It is hypothesized that FICD is a sensor of ATP/ADP ratios that outputs into regulation of BiP activity. This is a very interesting idea and a mechanism not previously seen, and it has implications for the regulation of other proteins by post-translational modifications.

The initial observation was that FICD knockout cells lost AMPylation, it could be restored weakly at low re-expression of FICD but not at high levels. The inhibitor mutant E234G promoted AMPylation in the cells, as did the L258D monomerization mutant. In vitro, three main assays were used: 1. radiolabeled total or net AMPylation of BiP; 2. release of fluorescent AMP to measure de-AMPylation; and 3. conversion of ATP to AMP in an AMPylation/de-AMPylation cycle. In addition, a catalytically inactive, disulfide-bonded trap FICD was constructed to protect AMP-BiP so AMPylation specifically could be measured. In these experiments, monomeric FICD had moderate total AMPylation, weaker de-AMPylation than WT, and robust AMPylation cycling, plus clear AMPylation in the presence of the trap. AMPylation by WT FICD increased at low concentrations, consistent with equilibrium shifting to monomers, but with the AMP trap AMPylation activity correlated directly with concentration. An engineered dimeric FICD had low AMPylation activity. In the previous structure, an H-bond network stabilized Glu234, and "relay" mutants (K256S, E242A) allowed AMPylation while still in dimers.

A series of crystallographic structures of WT, monomer and relay mutants in apo and ATP-bound states were solved. The monomer-ATP structures had the TPR domain in a reverse orientation, but the functional consequences are unclear. The catalytic FIC domain was the same except that WT with ATP bound no Mg ion, and the other ATP structures did. ATP in the WT structure was in an arrangement too far from the catalytic His and incompatible with AMPylation, in contrast to the mutant structures. In the mutants, Glu234 was shifted enough for activity, unlike in WT. Dimerization allosterically affected the active site, and it was proposed that ATP could also affect dimerization. Using BLI, FICD monomer mutants bound more BiP than dimers. ATP decreased the binding of BiP to monomers, and had less of an effect on dimers. ATP also promoted the dissociation of dimers into monomers, and ADP had less of an effect. It was proposed that FICD could react to the ATP/ADP ratio in cells, with high ATP levels favoring BiP AMPylation by monomers, and high ADP levels such as during stress favoring dimers and de-AMPylation to permit BiP function.

The experiments were conducted properly with appropriate controls. On the whole, the conclusion that AMPylation is activated by conversion from dimer to monomer is very well supported. A lot of work was put in to demonstrate how monomerization can activate AMPylation enzymatic activity. However it is not clear that ATP alone is responsible for the physiological conversion to dimers. The main concerns:

1. The weakest link is the mechanism that promotes monomerization. A single experiment Fig. 6E suggests that ATP promotes faster dissociation of dimers than ADP or no nucleotide. Fig. 6E and S8E are the same experiment presented with different vertical axes. The total association/dissociation is shown in S8E, and the change in dissociation with ATP is moderate. This is not sufficient to conclude that the equilibrium shifts to the monomer enough for substantial

amounts of AMPylation. The on rates are not measured so affinity KD cannot be estimated. Some other methods to show a change in the equilibrium amounts of monomers are needed. Perhaps protein footprinting of the dimerization interface? At the least, another independent method should confirm the dissociation of dimers with ATP.

2. How sensitive is FICD monomerization and AMPylation activity to differences in ATP levels? That is, what is the dynamic range of ATP/ADP ratio sensing? ADP is not an allosteric inhibitor, but a competitor for ATP, so at the minimum the response to ATP concentrations has to be addressed. I realize that the AMPylation enzymatic reaction rate will most likely also depend on ATP concentration. Maybe if AMPylation catalytic activity is less sensitive to changes in ATP concentration than dimer-monomer transition, WT and L258D AMPylation rates could be compared at increasing ATP concentrations. The WT FICD may start at lower rates than L258D but have a steeper response curve.

3. An alternative to point 2, would be an experiment showing how much monomer is present at different ATP concentrations. It would be an extension of experiments addressing point 1.

4. The other part of the hypothesis, is whether ATP concentrations in the ER lumen change enough during and after stress to activate FICD AMPylation in its sensitive range, points 2 and 3 above. Is it plausible that ATP concentrations or ATP/ADP ratios during recovery from stress (mimicked by CHX treatment) are higher than in normal steady state growth? An ER ATP sensor has been published: *Mol Biol Cell*. 2014, 25(3):368-79.

5. The effect of ATP on BiP binding and release by monomer FICD does not seem connected to the rest of the mechanism, except for the general idea of allostery. Ground state destabilization is mentioned but is speculative. BiP and AMP-BiP binding could be compared, perhaps FICD dimers will bind AMP-BiP better than BiP, and monomers vice versa.

6. More of the in vitro AMPylation reactions could be quantified and compared. WT, E234G, monomer and constitutive dimer reactions with and without trap are scattered over several panels, with different reaction conditions and times, and hard to compare directly. In the manuscript the results are mostly qualitative. It would be important to compare the L258D monomer with and without trap side by side with the WT and constitutive dimer. The monomer without trap is quite important as it addresses the central conclusion, but it is only in supplemental Fig. S4.

Referee #3:

Perera et al show that the protein FICD (de-)AMPylates BiP depending on its oligomeric state. This is a very exciting mechanistic finding. The same group has established before that FICD can regulate BiP activity both by AMPylation and de-AMPylation. However, it was unclear how the decision is made what FICD is doing to BiP. Understanding post-translation modification of molecular chaperones is an important topic to understand regulation of protein homeostasis.

The authors employ a wide range of methods to support their findings. The data are conclusive, of high quality and support the main message.

There are some minor issues the authors should address before publication.

1. The authors present structural data supporting the mode of action of FICD. This is only part of the story, the other part is BiP. The reader would be grateful for visualisation at atomic resolution how dimerisation affects interaction with BiP. The authors should combine their crystallographic findings for FICD with structural representation of BiP in open and closed conformation to make this point more clear.

2. The authors use human FICD and hamster BiP, according to the methods section. The authors should indicate already in the results section which proteins they use. Also, they should indicate to which extent residue numbers differ between human and hamster BiP. In addition, an alignment in supplementary could also be helpful.

Response to reviewers' comments on EMBOJ-2019-102177

N.B. our point-by-point responses to the reviewers' comments are provided in **orange**.

Referee #1:

The manuscript by Perera et al. provides several lines of evidence defining the basis for the switch in FICD's activity between AMPylation and deAMPylation of the ER chaperone BiP, critical for UPR. Specifically, the transition of FICD from a dimeric enzyme, that deAMPylates BiP, to a monomer with potent AMPylation activity is mainly demonstrated using relevant FICD variants in vitro and in cells. Experimental evidence fully supports the main conclusions.

One important aspect of the proposed model in Figure 7 is the identity of cellular regulators that affect FICD monomer/dimer transitions and in turn AMPylation/ deAMPylation. It is speculated that enzymatic activity might be regulated by the ADP/ATP ratio in the ER, based on limited data in vitro. It could be better to include in this discussion other possibilities (other changes in the ER environment, including other small molecules and interaction partners). It is understandable that further supporting data will be another major undertaking.

The revised version provides further experimental support for the conclusion that the FICD monomer-dimer equilibrium can be modulated by ADP/ATP (see response to Referee #2, Point 1). Nonetheless we recognise the validity of this reviewer's comment concerning the potential role of other factors in regulating BiP AMPylation and acknowledge that FICD's ability to sense a changing ADP/ATP ratio in the ER may be but one component of the regulatory mechanism for BiP AMPylation/deAMPylation. As such we have now included explicit references to other possible modes of regulation, in *Discussion: Paragraph 8*, which may also facilitate a link between ER unfolded protein load level and the level of AMPylated BiP (e.g. molecular crowding effects and FICD-substrate availability). We thank the reviewer for highlighting this point.

Referee #2:

The manuscript addresses the mechanisms that determine whether BiP is ampylated or de-ampylated by FICD. Ampylation is inhibitory of BiP and occurs under specific conditions, such as recovery from stress. FICD is a homodimer that de-ampylates BiP, and the ampilation reaction by FICD is observed when an inhibitory Glu234 is mutated. The manuscript concludes that monomerization of FICD allows ampilation activity, by changing an allosteric network of bonds that position Glu234. High resolution structures suggest that dimerization affects ATP binding, and ATP in turn is found to affect monomerization and BiP binding. It is hypothesized that FICD is a sensor of ATP/ADP ratios that outputs into regulation of BiP activity. This is a very interesting idea and a mechanism not previously seen, and it has implications for the regulation of other proteins by post-translational modifications.

The initial observation was that FICD knockout cells lost AMPylation, it could be restored weakly at low re-expression of FICD but not at high levels. The inhibitor mutant E234G promoted AMPylation in the cells, as did the L258D monomerization mutant. In vitro, three main assays were used: 1. radiolabeled total or net AMPylation of BiP; 2. release of fluorescent AMP to measure de-AMPylation; and 3. conversion of ATP to AMP in an AMPylation/de-AMPylation cycle. In addition, a catalytically inactive, disulfide-bonded trap FICD was constructed to protect AMP-BiP so AMPylation specifically could be measured. In these experiments, monomeric FICD had moderate total AMPylation, weaker de-AMPylation than WT, and robust AMPylation cycling, plus clear AMPylation in the presence of the trap. AMPylation by WT FICD increased at low concentrations, consistent with equilibrium shifting to monomers, but with the AMP trap AMPylation activity correlated directly with concentration. An engineered dimeric FICD had low AMPylation activity. In the previous structure, an H-bond network stabilized Glu234, and "relay" mutants (K256S, E242A) allowed AMPylation while still in dimers.

A series of crystallographic structures of WT, monomer and relay mutants in apo and ATP-bound

states were solved. The monomer-ATP structures had the TPR domain in a reverse orientation, but the functional consequences are unclear. The catalytic FIC domain was the same except that WT with ATP bound no Mg ion, and the other ATP structures did. ATP in the WT structure was in an arrangement too far from the catalytic His and incompatible with AMPylation, in contrast to the mutant structures. In the mutants, Glu234 was shifted enough for activity, unlike in WT. Dimerization allosterically affected the active site, and it was proposed that ATP could also affect dimerization. Using BLI, FICD monomer mutants bound more BiP than dimers. ATP decreased the binding of BiP to monomers, and had less of an effect on dimers. ATP also promoted the dissociation of dimers into monomers, and ADP had less of an effect. It was proposed that FICD could react to the ATP/ADP ratio in cells, with high ATP levels favoring BiP AMPylation by monomers, and high ADP levels such as during stress favoring dimers and de-AMPylation to permit BiP function.

The experiments were conducted properly with appropriate controls. On the whole, the conclusion that AMPylation is activated by conversion from dimer to monomer is very well supported. A lot of work was put in to demonstrate how monomerization can activate AMPylation enzymatic activity. However it is not clear that ATP alone is responsible for the physiological conversion to dimers. The main concerns:

We thank the reviewer for the care with which they read our manuscript and for their thoughtful and constructive critiques.

1. The weakest link is the mechanism that promotes monomerization. A single experiment Fig. 6E suggests that ATP promotes faster dissociation of dimers than ADP or no nucleotide. Fig. 6E and S8E are the same experiment presented with different vertical axes. The total association/dissociation is shown in S8E, and the change in dissociation with ATP is moderate. This is not sufficient to conclude that the equilibrium shifts to the monomer enough for substantial amounts of AMPylation. The on rates are not measured so affinity K_D cannot be estimated. Some other methods to show a change in the equilibrium amounts of monomers are needed. Perhaps protein footprinting of the dimerization interface? At the least, another independent method should confirm the dissociation of dimers with ATP.

In response to the reviewer's legitimate critique we now present (in Figure 6F-G) an orthogonal assay for measuring changes in the oligomeric-state of FICD in vitro. To achieve this we have introduced a cysteine residue onto the surface of FICD that once decorated with a fluorophore gives rise to a FRET probe sensitive to FICD's oligomeric state. The fluorescent signal measured under conditions in which the probe is fully engaged in dimerisation with other labelled FICD molecules (high FRET) or other unlabelled FICD molecules (no FRET) provided an optical readout for the monomer-dimer equilibrium and enabled measurement of the effect of ATP/ADP on this parameter at steady-state. The new findings (derived by an assay that is orthogonal to the BLI measurements of the effect of nucleotide on the dissociation kinetics of the FICD dimer) substantiates the conclusion that an increasing ADP/ATP ratio favours FICD dimerisation by affecting the monomer-dimer equilibrium at steady state.

2. How sensitive is FICD monomerization and AMPylation activity to differences in ATP levels? That is, what is the dynamic range of ATP/ADP ratio sensing? ADP is not an allosteric inhibitor, but a competitor for ATP, so at the minimum the response to ATP concentrations has to be addressed. I realize that the AMPylation enzymatic reaction rate will most likely also depend on ATP concentration. Maybe if AMPylation catalytic activity is less sensitive to changes in ATP concentration than dimer-monomer transition, WT and L258D AMPylation rates could be compared at increasing ATP concentrations. The WT FICD may start at lower rates than L258D but have a steeper response curve.

We believe these valid concerns are addressed by the new FRET assay, see response to point 1.

3. An alternative to point 2, would be an experiment showing how much monomer is present at different ATP concentrations. It would be an extension of experiments addressing point 1.

See response to point 1.

4. The other part of the hypothesis, is whether ATP concentrations in the ER lumen change enough during and after stress to activate FICD AMPylation in its sensitive range, points 2 and 3 above. Is it plausible that ATP concentrations or ATP/ADP ratios during recovery from stress (mimicked by CHX treatment) are higher than in normal steady state growth? An ER ATP sensor has been published: Mol Biol Cell. 2014, 25(3):368-79.

We agree that measurement of changes in ER luminal ATP/ADP ratio in cells exposed to diverse stress conditions is an important goal for the future and we are monitoring the state of development of probes suited for such measurements closely. To our knowledge, at present such a ratiometric probe (PercevalHR) has only been developed for the cytosol (Tantama *et al.*, 2013). However, we believe that addressing this issue with the rigor and thoroughness it deserves is an endeavour that lies outside the scope of this paper.

5. The effect of ATP on BiP binding and release by monomer FICD does not seem connected to the rest of the mechanism, except for the general idea of allostery. Ground state destabilization is mentioned but is speculative. BiP and AMP-BiP binding could be compared, perhaps FICD dimers will bind AMP-BiP better than BiP, and monomers vice versa.

In Figure 5 we illustrate the distinct differences in ATP binding mode observed, crystallographically, between monomeric and dimeric FICD. We therefore considered analysis of the effect of nucleotide at a more holistic level to be a logical experimental follow-up (Figure 6). The resulting differential effect of nucleotide on monomeric vs dimeric FICD pre-AMPylation complex kinetics, as pointed out by the reviewer, not only demonstrates the allostery mediated by nucleotide (which was hitherto not apparent) but also suggests another means by which monomerisation may facilitate BiP-AMPylation; that is to say, monomerisation might sensitise FICD to an allosteric form of enzyme-(co)substrate destabilisation (which would increase the k_{cat} for AMPylation). The discussion of the revised version of the paper now emphasises that this is an unproven speculation, albeit one that we find plausible, informative and therefore worthy of sharing with our readers.

The reviewer's reasonable suggestion that we compare the binding of AMPylated and unmodified BiP to monomeric and dimeric FICD is addressed in Figure S2G. The reviewer's hypothesis was indeed confirmed by this experiment. In the revised manuscript these findings are now emphasised and referenced with respect to Figure 6A-B in *Results: ATP is an allosteric modulator of FICD: Paragraph 2*: "In contrast, with respect to pre-deAMPylation complex assembly, AMPylated BiP bound more tightly to dimeric FICD^{H363A} than to monomeric FICD^{L258D-H363A} (Figure S2G)." We believe these changes, made at the reviewer's behest, clarify this issue.

6. More of the in vitro AMPylation reactions could be quantified and compared. WT, E234G, monomer and constitutive dimer reactions with and without trap are scattered over several panels, with different reaction conditions and times, and hard to compare directly. In the manuscript the results are mostly qualitative. It would be important to compare the L258D monomer with and without trap side by side with the WT and constitutive dimer. The monomer without trap is quite important as it addresses the central conclusion, but it is only in supplemental Fig. S4.

The large amount of information communicated in this paper lends itself to many permutations in terms of the arrangement of the data panels. We believe that this is principally an editorial issue and that the order provided in the manuscript in its present form is well suited to its narrative. In deference to the reviewer, we have now quantified the BiP-AMP signals in Figure 1F to compare directly the signal generated by wild-type FICD and the monomeric mutants; based on the representative experiment shown and two additional independent repeats (see new bar graph next to gels in Figure 1F). However, it is worth noting that the time-dependent changes in the abundance of AMPylated BiP in these gel based experiments represent a composite of two antagonistic biochemical activities (AMPylation and deAMPylation) and thus their quantification is less informative than the more complete quantitative description of the wild-type and monomeric FICDs activities provided in Figure 2 (total activity, AMPylation and deAMPylation). Figure 2C compares AMP production – the most sensitive and quantitative readout for net enzymatic activity (AMPylation and deAMPyation). Figure 2A and S2A quantitatively address de-AMPylation activities at two different enzyme concentrations. Figure 2E compares directly the AMPylation activities of wild-type and monomeric FICD in presence of the trap. The same workflow was

conducted for other mutants introduced in later figures and we included in each experiment wild-type plus FICD^{E234G} or FICD^{L258D} as a reference to allow comparison between panels and figures.

Referee #3:

Perera et al show that the protein FICD (de-)AMPylates BiP depending on its oligomeric state. This is a very exciting mechanistic finding. The same group has established before that FICD can regulate BiP activity both by AMPylation and de-AMPylation. However, it was unclear how the decision is made what FICD is doing to BiP. Understanding post-translation modification of molecular chaperones is an important topic to understand regulation of protein homeostasis.

The authors employ a wide range of methods to support their findings. The data are conclusive, of high quality and support the main message.

There are some minor issues the authors should address before publication.

1. The authors present structural data supporting the mode of action of FICD. This is only part of the story, the other part is BiP. The reader would be grateful for visualisation at atomic resolution how dimerisation affects interaction with BiP. The authors should combine their crystallographic findings for FICD with structural representation of BiP in open and closed conformation to make this point more clear.

We agree with the reviewer on the value of atomic resolution information (i.e. a co-complex structure) of FICD interacting with BiP. This is the subject of ongoing work in our lab (and perhaps other labs too). We know that the AMPylation substrate is BiP:ATP (Preissler *et al*, 2015) and that the substrate for deAMPylation is BiP-AMP [which is locked in a BiP:ATP-like conformation (Preissler *et al*, 2017)], thus we anticipate that both BiP and BiP-AMP will interact with FICD in a lid open, NBD-SBD domain docked, Hsp70:ATP-like conformation. However, at this time we cannot provide a more detailed model of the BiP-FICD complex beyond this rough sketch and therefore cannot responsibly provide the visual schema of the reaction intermediates that the reviewer would like to see.

2. The authors use human FICD and hamster BiP, according to the methods section. The authors should indicate already in the results section which proteins they use. Also, they should indicate to which extent residue numbers differ between human and hamster BiP. In addition, an alignment in supplementary could also be helpful.

Apart from their signal sequences, which are cleaved in the mature protein, Chinese hamster and human BiP are essentially identical. From residues 20-654 there is only one residue that varies Ala650Ser (going from human to hamster). This amino acid is in the unstructured C-terminus of the protein, two amino acids upstream of the terminal KDEL sequence. The use of human FICD and hamster BiP together, therefore, effectively represents a homologous human system. In the revised manuscript, we have made reference to this evolutionary conservation in *Methods: Protein purification: BiP: Paragraph 1*. We thank the reviewer for allowing us to clarify this point.

References

- Preissler S, Rato C, Chen R, Antrobus R, Ding S, Fearnley IM & Ron D (2015) AMPylation matches BiP activity to client protein load in the endoplasmic reticulum. *Elife* **4**: e12621
- Preissler S, Rohland L, Yan Y, Chen R, Read RJ & Ron D (2017) AMPylation targets the rate-limiting step of BiP's ATPase cycle for its functional inactivation. *Elife* **6**: e29428
- Tantama M, Martínez-François JR, Mongeon R & Yellen G (2013) Imaging energy status in live cells with a fluorescent biosensor of the intracellular ATP-to-ADP ratio. *Nat. Commun.* **4**: 2550

Thank you for submitting your revised manuscript for our consideration. It has now been seen once more by two of the original referees (see comments below). I am pleased to say that the referees overall find that their comments have been satisfactorily addressed. We will therefore be happy to accept the study for publication, after a final minor revision to address several editorial issues that are listed in detail below.

In addition, you will see that referee #3 notes that the impact of the study would be increased by using homology modeling to more directly answer her/his first minor concern regarding the mapping of interactions on the 3D structure. While we would not insist on this remaining point at this stage, we still encourage you to consider this aspect and at least carefully respond to the comment.

REFEREE REPORTS

Referee #2:

The new FRET data in Fig. 6G shows that the population shifts to the monomer with ATP, with a midpoint of around 0.1 mM ATP and a plateau above 1 mM. ADP over a similar concentration range shifts toward the dimer. When ATP is kept constant at 5 mM, increasing ADP shifts the enzyme from monomer to dimer, and the midpoint is around 0.2 mM ADP. Although the shifts are not large, from 0.28 to 0.35 fraction of monomer, the nucleotide concentrations are in the physiologic range, and the ratio of 5 mM ATP to 0.2 mM ADP is close to what are thought to be normal values. Real measurements of ER ATP and ADP under stress would further establish the mechanism.

The difference between BiP and BiP-AMP binding by dimer and monomer is demonstrated clearly.

Although I would still prefer to see more quantifications of the AMPylations, I understand the author's argument that AMP production is a better overall measure.

Referee #3:

The authors satisfied the second minor point of reviewer #3. They did not adequately address the first minor point, mapping the interactions on the 3D structure of BiP. The authors argue that there is no BiP structure available yet. However, there are plenty of Hsp70 structures, which allows homology modelling for BiP. While this is not the same as a high resolutions structure, it provides the overall orientation of binding sites etc, which will be very helpful to convey the message to readers. This is an important study that is also relevant to colleagues more interested in cytosolic Hsp70 chaperones, so such homology models would increase the impact for the chaperone filed. It would be a pity if the authors miss this opportunity.

EMBOJ-2019-102177R

Point-by-point response to the reviewer's comments:

We are pleased by the favorable assessment of our revised manuscript and by the invitation to submit a final version suitable for publication.

Below please find the reviewer's comments in black ink and our response in purple ink

Referee #2:

The new FRET data in Fig. 6G shows that the population shifts to the monomer with ATP, with a midpoint of around 0.1 mM ATP and a plateau above 1 mM. ADP over a similar concentration range shifts toward the dimer. When ATP is kept constant at 5 mM, increasing ADP shifts the enzyme from monomer to dimer, and the midpoint is around 0.2 mM ADP. Although the shifts are not large, from 0.28 to 0.35 fraction of monomer, the nucleotide concentrations are in the physiologic range, and the ratio of 5 mM ATP to 0.2 mM ADP is close to what are thought to be normal values. Real measurements of ER ATP and ADP under stress would further establish the mechanism.

The difference between BiP and BiP-AMP binding by dimer and monomer is demonstrated clearly.

Although I would still prefer to see more quantifications of the AMPylations, I understand the author's argument that AMP production is a better overall measure.

We are pleased by this assessment.

We agree that measurements of ATP, ADP and their ratios in the ER under various physiological conditions are an important frontier for this research and we hope to contribute to this line of inquiry in the future.

Referee #3:

The authors satisfied the second minor point of reviewer #3. They did not adequately address the first minor point, mapping the interactions on the 3D structure of BiP. The authors argue that there is no BiP structure available yet. However, there are plenty of Hsp70 structures, which allows homology modelling for BiP. While this is not the same as a high resolutions structure, it provides the overall orientation of binding sites etc, which will be very helpful to convey the message to readers. This is an important study that is also relevant to colleagues more interested in cytosolic Hsp70 chaperones, so such homology models would increase the impact for the chaperone filed. It would be a pity if the authors miss this opportunity.

We thank the reviewer for their sincere effort to advise us how to improve the message.

We have considered the matter carefully and wish to offer the following response:

There are published high resolution structures of BiP in its ATP state, AMPylated state and ADP state. Likewise, there are high resolution structures of FICD. The difficulty lies in providing a scientifically valid model of the substrate (BiP) docked onto the enzyme (FICD) in the absence of any structural information on the complex. The only published data relating to the 'interaction surface' of BiP and FICD is that Thr518 of BiP (the residue which becomes AMPylated) needs to be in proximity to His363 of FICD for catalysis (the general base in FICD-mediated AMPylation). Given this scarcity of structural constraints molecular docking of FICD and BiP will doubtlessly provide a number of plausible solutions. The utility and indeed validity of displaying the computationally 'most probable' solution is questionable without extensive experimental testing, which we believe falls outside the scope of this manuscript. Nevertheless, such a molecular docking has been carried out by the Mattoo group (Sanyal et al., 2018); Figure 6 therein presents one plausible structure of a 2:2 heterotetrameric FICD:BiP complex docked in a conformation which would permit Thr518 AMPylation.

References

- Sanyal, A., Zbornik, E. A., Watson, B. G., Christoffer, C., Ma, J., Kihara, D., & Mattoo, S. (2018). Kinetic And Structural Parameters Governing Fic-Mediated Adenylylation/AMPylation of the Hsp70 chaperone, BiP/GRP78. *BioRxiv*, 494930. <https://doi.org/10.1101/494930>

3rd Editorial Decision

19th July 2019

Thank you for submitting your revised manuscript for our consideration. Prior to formal acceptance and transferring of your manuscript to our publisher there are some minor editorial issues that need to be addressed.

YOU MUST COMPLETE ALL CELLS WITH A PINK BACKGROUND ↓
PLEASE NOTE THAT THIS CHECKLIST WILL BE PUBLISHED ALONGSIDE YOUR PAPER

Corresponding Author Name: Steffen Preissler & David Ron
Journal Submitted to: EMBO Journal
Manuscript Number: EMBOJ-2019-102177